# Striatal dopamine dissociates methylphenidate effects on value-based versus surprise-based reversal learning

Ruben van den Bosch [1] ✉, Britt Lambregts[2], Jessica Määttä [3], Lieke Hofmans[4], Danae Papadopetraki[2], Andrew Westbrook [5], Robbert-Jan Verkes[2], Jan Booij[6,7] & Roshan Cools [2]

Psychostimulants such as methylphenidate are widely used for their cognitive enhancing effects, but there is large variability in the direction and extent of these effects. We tested the hypothesis that methylphenidate enhances or impairs reward/punishment-based reversal learning depending on baseline striatal dopamine levels and corticostriatal gating of reward/punishment-related representations in stimulus-specific sensory cortex. Young healthy adults (N = 100) were scanned with functional magnetic resonance imaging during a reward/punishment reversal learning task, after intake of methylphenidate or the selective $D_{2/3}$-receptor antagonist sulpiride. Striatal dopamine synthesis capacity was indexed with [$^{18}$F]DOPA positron emission tomography. Methylphenidate improved and sulpiride decreased overall accuracy and response speed. Both drugs boosted reward versus punishment learning signals to a greater degree in participants with higher dopamine synthesis capacity. By contrast, striatal and stimulus-specific sensory surprise signals were boosted in participants with lower dopamine synthesis. These results unravel the mechanisms by which methylphenidate gates both attention and reward learning.

Adaptive behavior requires flexible updating of our actions and predictions in response to the constant changes around us. Catecholamine transmission is well-accepted to be essential for such flexible behavior, as demonstrated by accumulating evidence from experimental psychopharmacology showing effects of catecholaminergic drugs on performance on cognitive tasks, including reversal learning, cognitive action gating and attention shifting[1–9]. Methylphenidate, a dopamine and noradrenaline transporter blocker, is the first-line pharmacological treatment for attention deficit hyperactivity disorder (ADHD), which is characterized by cognitive deficits[10,11]. In addition, it

is widely used by healthy people as a psychostimulant for its cognition-enhancing effects[12–15]. However, there is large interindividual variability in the direction of the effects[2,16,17] and the mechanisms underlying these effects remain unclear. This poses a major problem for treatment strategies in psychiatry and raises questions about the use of methylphenidate as a therapeutic or a smart drug.

Here, we investigate the neural mechanisms of methylphenidate's effects on reward-related reversal learning: the ability to flexibly adapt behavior based on changes in reward and punishment contingencies. While evidence from work with nonhuman animals suggests that many

[1]Radboud University, Donders Institute for Brain, Cognition and Behaviour, Nijmegen, The Netherlands. [2]Radboud University Medical Center, Department of Psychiatry, Donders Institute for Brain, Cognition and Behaviour, Nijmegen, The Netherlands. [3]Department of Clinical Neuroscience, Karolinska Institutet, Stockholm, Sweden. [4]Department of Developmental Psychology, University of Amsterdam, Amsterdam, The Netherlands. [5]Cognitive, Linguistic & Psychological Sciences Department, Brown University, Providence, RI, USA. [6]Department of Radiology and Nuclear Medicine, Amsterdam University Medical Centers, location Academic Medical Center, Amsterdam, The Netherlands. [7]Radboud University Medical Center, Department of Medical Imaging, Nijmegen, The Netherlands. ✉e-mail: ruben.vandenbosch@donders.ru.nl

of methylphenidate's cognitive enhancing effects involve direct action in the prefrontal cortex[18], work with human volunteers has suggested a key role for dopamine in the striatum[2,19]. A large body of evidence indicates that both the therapeutic effects[20,21] and potential for abuse[22] of methylphenidate are related to its effects on reinforcement as well as surprise[23–26], and associated striatal dopamine signaling[19].

A causal role for striatal dopamine in reward-based reversal learning is substantiated by a large body of evidence from studies with nonhuman primates[27], experimental rodents[28,29], and patients with Parkinson's disease[30,31]. Studies in young healthy volunteers have shown that administration of methylphenidate modulates reversal-related BOLD signal in the striatum[3] and impairs reversal learning in proportion to the degree that methylphenidate increased striatal dopamine release[2]. Genetic variation in a common dopamine transporter polymorphism, affecting primarily striatal dopamine, predicted increased perseveration during reversal learning due to increased reliance on prior reinforcement[32]. This evidence generally concurs with the observation that injection of D-amphetamine in the striatum of rats potentiates behavioral control by stimuli formerly associated with reward (i.e., conditioned reinforcement) in a dopamine-dependent manner[33]. While these studies establish a causal role for striatal dopamine specifically in reward-driven compulsivity as measured with reversal learning, the corticostriatal mechanisms underlying the large interindividual variability in methylphenidate's effects remain unclear.

According to influential neurocomputational modeling work[34], striatal dopamine flexibly gates task-relevant actions and representations maintained in the cortex by promoting activity in the direct (Go) versus indirect (NoGo) pathway of the basal ganglia in proportion to reward (versus punishment) prediction error. Inspired by this work, we hypothesized that methylphenidate boosts reversal learning by acting on striatal dopamine to gate attention to reward-associated cortical representations. To test this output gating hypothesis[35,36], we index drug effects on neural signals not only in the striatum and the strongly connected prefrontal brain regions, but also in task-relevant cortical areas that are specialized for the processing of selective stimulus categories. Specifically, we measured the relative stimulus-specificity of visual association cortex responses to face and scene stimuli[37–39] while participants learned and updated their association with reward and punishment. We predicted that methylphenidate boosts reversal signals in the striatum, reward versus punishment reversal-learning performance, and the associated reward versus punishment reversal signals in the stimulus-specific visual association cortex.

To address the role of striatal dopamine in these effects, we compared the effects of the nonspecific catecholamine enhancer methylphenidate with those of the selective dopamine $D_{2/3}$-receptor antagonist sulpiride. At low doses, sulpiride acts preferentially presynaptically to increase synaptic dopamine levels in the striatum[34,40,41]. Therefore, we predicted parallel effects of methylphenidate and sulpiride (preregistration: https://osf.io/ey4j7/). To further establish the dependency of the effects of methylphenidate and sulpiride on striatal dopamine, we additionally measured each participant's striatal dopamine synthesis capacity directly, with [18F]DOPA positron emission tomography (PET). We focused on the cognitive subregion of the striatum, the caudate nucleus, previously demonstrated to be essential for reversal learning[27,42], but additionally explored the role of dopamine in the motor and motivational subregions (putamen and nucleus accumbens, respectively)[43,44].

The PET design also enabled us to test the pervasive baseline-dependency hypothesis of dopaminergic drugs, which predicts that methylphenidate's effects depend on baseline levels of striatal dopamine[45] (https://osf.io/ey4j7/). Prior PET studies focusing on dopamine suggested a nonlinear, inverted-U-shaped relationship between dopamine and cognitive performance[45], with participants with low dopamine synthesis capacity benefiting from dopaminergic drug administration, while "high-dopamine participants" were

impaired by the same dopaminergic drug[46]. However, previous PET studies focusing on dopamine have been conducted only in small samples[7,9,46,47], precluding inferences about the reliability of observed between-subject variability. This study combined pharmacological functional magnetic resonance imaging (fMRI) with [18F]DOPA PET in 100 healthy volunteers, thus providing a large sample to study inter-individual differences in the cognitive and neural response to methylphenidate.

## Results

A deterministic reversal learning task with face and scene stimuli was employed, designed to separately assess reward-based and punishment-based reversal learning (Fig. 1a)[48]. Across all three sessions, unexpected reward and punishment outcomes, which signaled reversals, strongly activated frontostriatal circuitry (Fig. 1b), improved accuracy and response times (RTs; Supplementary Results; Supplementary Fig. 1), and increased face/scene stimulus-specific BOLD signal in visual association cortex, consistent with the need for updating outcome predictions for faces/scenes following unexpected events (Fig. 1c).

Task performance was highly sensitive to both methylphenidate and sulpiride (all reported drug effects reflect a comparison of drug minus placebo). Consistent with its established performance-enhancing effects, methylphenidate increased overall task accuracy and shortened overall RTs ($N = 88$; Fig. 1d, e; Supplementary Table 1; main drug effect on accuracy in Bayesian mixed-effects model: estimate [B] = 0.268, 95% credible interval [CI] = [0.184, 0.354]; main drug effect on RT: B = −0.016, CI = [−0.028, −0.004]). In contrast, sulpiride decreased overall accuracy and did not significantly affect overall RTs (accuracy: B = −0.134, CI = [−0.203, −0.065]; RT: B = −0.003, CI = [−0.013, 0.006]).

The main findings described below are summarized in a schematic figure in "Discussion".

### Methylphenidate enhances striatal BOLD signal

Methylphenidate boosted outcome-related activity in the striatum. Specifically, it increased BOLD signal during both unexpected rewards and punishments in the bilateral ventral putamen, with additional clusters in the bilateral posterior putamen ($N = 85$; Fig. 2a; peak voxel ventral putamen: $x, y, z = −30, 3, −5, Z = 4.34$, $p_{peak\ FWE\ SVC} = 0.005$; peak voxel posterior putamen: $x, y, z = 29, −13, 7, Z = 4.51$, $p_{peak\ FWE\ SVC} = 0.002$).

As anticipated, there was great variability between participants, with methylphenidate boosting signals in the caudate nucleus to a greater degree in participants with lower dopamine synthesis capacity in the caudate nucleus, as indexed by [18F]DOPA influx ($k_i^{cer}$; Fig. 2c, d; Supplementary Fig. 3a; peak voxel: $x, y, z = 10, 10, 7, Z = 3.83$, $p_{peak\ FWE\ SVC} = 0.046$). The key role of dopamine in these effects was further substantiated by a parallel effect of sulpiride, which also boosted reversal-related caudate BOLD signal depending on caudate dopamine synthesis capacity ($N = 82$; Fig. 2c, e; Supplementary Fig. 3b; sulpiride x expectancy x caudate nucleus $k_i^{cer}$ peak voxel: $x, y, z = 13, 6, 13, Z = 4.04$, $p_{peak\ FWE\ SVC} = 0.020$). Neither drug effect depended on dopamine synthesis capacity in the putamen or nucleus accumbens, and there were no effects of sulpiride on BOLD signal to unexpected outcomes when dopamine synthesis capacity was not taken into account.

### Striatal dopamine boosts stimulus specificity in the visual cortex

To investigate the hypothesis that striatal dopamine gates currently relevant cortical representations, we assessed how methylphenidate, sulpiride, and striatal dopamine synthesis capacity affected activity in the task-relevant stimulus-specific visual association cortices. To this end, we leveraged the known face versus scene-specificity of activity in the fusiform face area (FFA) versus parahippocampal place area (PPA), respectively, which was particularly pronounced when participants

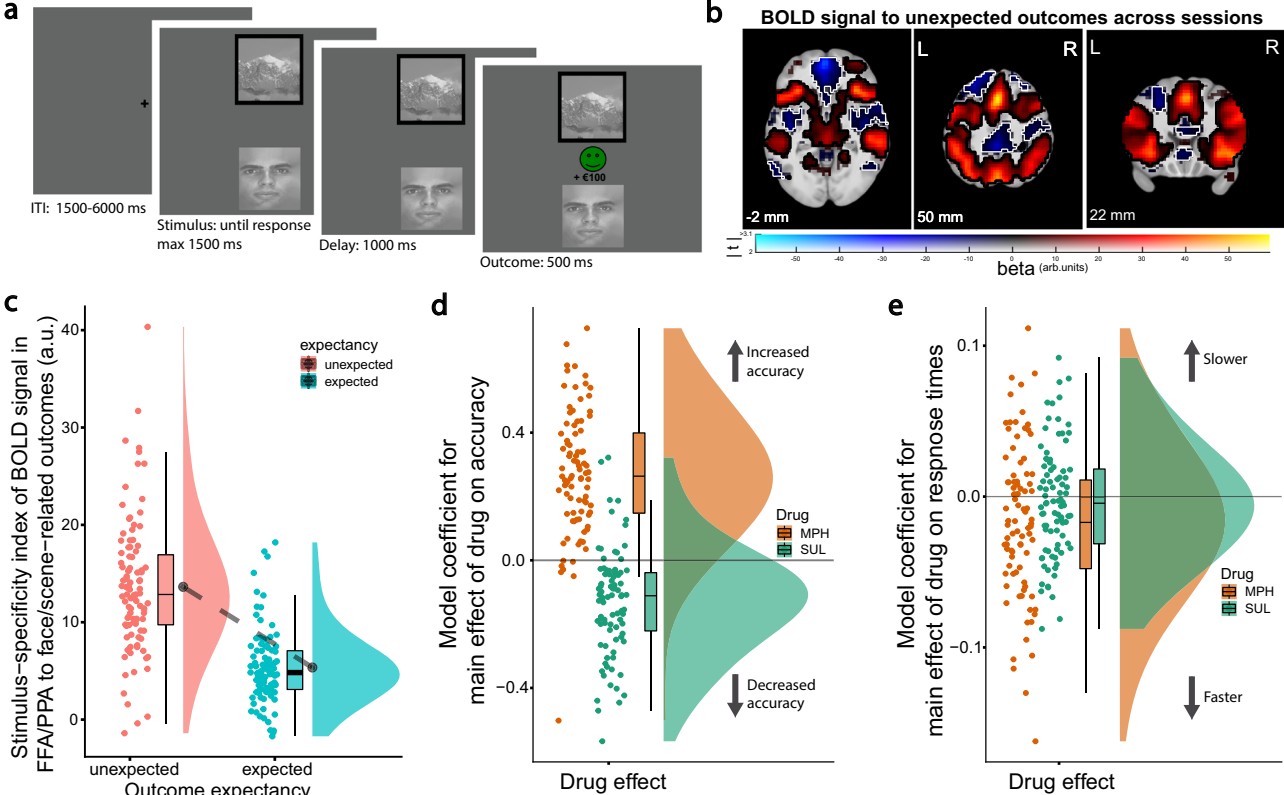

**Fig. 1 | Task and main effects. a** Reversal-learning paradigm. Two stimuli, a face and a scene, were presented simultaneously. One was associated with a reward outcome, the other with a punishment outcome, with 100% deterministic contingencies. On each trial the computer selected one image, and the participant's task was to predict whether the highlighted stimulus would be followed by a reward or punishment outcome. Then the actual outcome was presented. The stimulus-outcome associations reversed regularly, which was signaled by either an unexpected reward or unexpected punishment outcome. Accuracy on trials immediately after an unexpected outcome (reversal trials) was the performance measure of interest. Task images were obtained with permission from ref. 48. **b** fMRI BOLD signal to unexpected versus expected outcomes collapsed across all three sessions (N = 94 participants). In these dual-coded images, color indicates the size of the contrast estimate and opacity codes the height of the *t* values (plotting procedure: refs. 113, 114). Voxels with *t* values above the threshold of *P* < 0.001, uncorrected,

are fully opaque. Significant clusters (here, *P* < 0.05 after whole-brain cluster-level family-wise error correction) are encircled in black for red blobs or in white for blue blobs. The results are overlaid on the group-average T1-weighted anatomical scan in MNI152 coordinate space. **c** Across all three sessions, unexpected outcomes increased face/scene stimulus-specific BOLD signal in visual association cortex (N = 94 participants; main effect of expectancy in ANOVA: $F_{(1,93)} = 182.97$, *P* = 2.2e-16). The stimulus-specificity index represents the outcome-related BOLD signal in the contrast (FFA: faces − scenes) − (PPA: faces − scenes). Boxplots show the median and 25th and 75th percentiles, with the whiskers extending max. 1.5 * interquartile range. Round dots next to the data density kernel represent the mean value. **d**, **e** Bayesian mixed-effects model coefficients for the main effects of methylphenidate and sulpiride (relative to placebo) on accuracy and response times (N = 88 participants). Boxplots defined as in panel **c**. Source data are provided with this paper. MPH methylphenidate, SUL sulpiride, arb units arbitrary units.

update outcome predictions for faces versus scenes (during unexpected outcomes; Fig. 1c). We reasoned that if striatal dopamine gates the activity in these regions in proportion to outcome surprise, then the dopaminergic drugs should affect this stimulus-specific reversal signal in visual cortex in a striatal dopamine synthesis-dependent manner.

Both methylphenidate and sulpiride increased stimulus-specific activity during unexpected outcomes to a greater degree in participants with lower dopamine synthesis capacity in the caudate nucleus (Fig. 3 and Supplementary Fig. 5). Thus, for "lower-dopamine participants", methylphenidate and sulpiride enhanced reversal-related signal in the FFA (compared with PPA) when an unexpected outcome was associated with the face stimulus (compared with the scene). Conversely, the drugs enhanced signal in the PPA (compared with FFA) when the unexpected outcome was associated with the scene stimulus (compared with the face). As was the case for the drug effects on striatal outcome surprise signals, the effects on visual cortex did not significantly differ with the valence of the unexpected outcome (methylphenidate × expectancy × valence × caudate nucleus $k_i^{cer}$: $F_{(1,83)} = 0.03$, *P* = 0.869; sulpiride × expectancy × valence × caudate nucleus $k_i^{cer}$: $F_{(1,80)} = 0.50$, *P* = 0.482). The drug effects did not vary as a function of dopamine synthesis capacity in the putamen or nucleus accumbens.

The effect of methylphenidate on stimulus specificity in visual cortex across participants correlated with its effect on outcome surprise signal in the caudate nucleus in a between-participants analysis (ρ = 0.223, *P* = 0.04; correlation with sulpiride's effects: ρ = 0.002, *P* = 0.982). Under placebo, the stimulus-specific reversal-related signal in FFA/PPA was greater for participants with higher dopamine synthesis capacity in the caudate nucleus (expectancy × caudate nucleus $k_i^{cer}$ for placebo only: $F_{(1,80)} = 4.19$, *P* = 0.044). In a supplementary psychophysiological interaction (PPI) analysis, we tested the relationship between caudate nucleus signal and stimulus specificity of FFA/PPA signal in a within-participant manner. The analysis revealed that sulpiride, but not methylphenidate (compared with placebo), increased functional connectivity between the right caudate nucleus and the FFA/PPA during unexpected outcomes for participants with lower dopamine synthesis capacity in the caudate nucleus (Supplementary Fig. 6).

**Striatal dopamine effects on prefrontal BOLD signal**
There were no effects of methylphenidate or sulpiride on task-related BOLD signal in the prefrontal cortex, also not when accounting for dopamine synthesis capacity in the caudate nucleus. However, exploratory whole-brain analyses as a function of

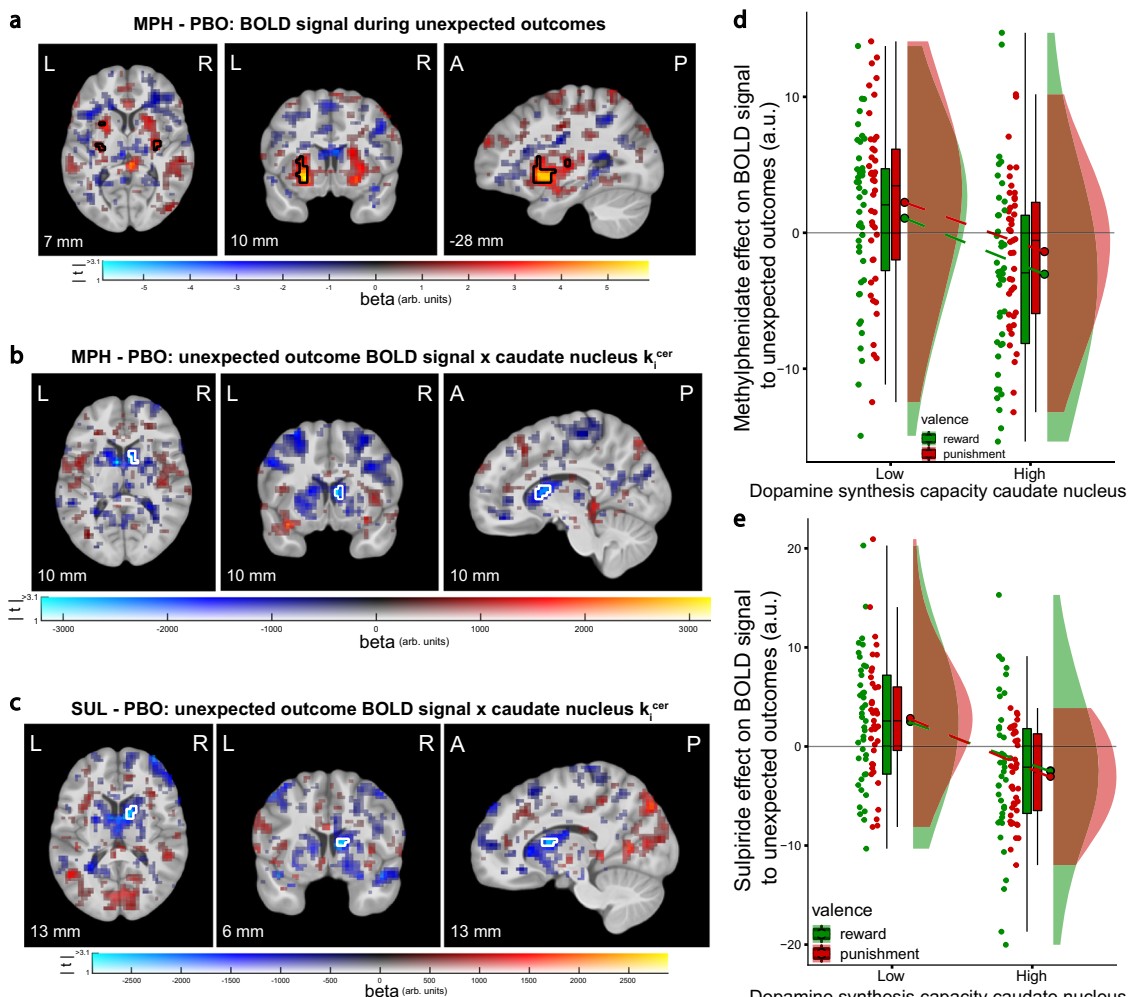

**Fig. 2 | Effects of methylphenidate and sulpiride (relative to placebo) on striatal BOLD signal to unexpected outcomes. a** Methylphenidate increased reversal-related striatal signals. Results of univariate regression analysis showing the effect of methylphenidate on BOLD signal related to unexpected outcomes (MPH − PBO: (unexpected reward − expected reward) + (unexpected punishment − expected punishment)). In the left putamen the activations formed one larger cluster that was also significant at the whole-brain level (peak voxel: $x, y, z = -30, 3, -5, Z = 4.34$, $p_{cluster\ FWE\ WB} < 0.001$). **b, c** Methylphenidate (**b**) and sulpiride (**c**) increased reversal-related BOLD signal in the right caudate nucleus to a greater degree in participants with lower caudate nucleus $k_i^{cer}$. Results of the effects of methylphenidate and sulpiride on the contrast for unexpected versus expected outcomes with caudate nucleus $k_i^{cer}$ as covariate. **d, e** Effect of methylphenidate (**d**) and sulpiride (**e**) on reversal-related BOLD signal in the significant clusters in **b** and **c**, respectively, as a function of low and high caudate nucleus $k_i^{cer}$, displayed separately for unexpected reward and punishment outcomes (median split for visualization). Panels **a,b,d**: $N = 85$ participants; panels **c,e**: $N = 82$. Figure conventions for panels **a–c** are as in Fig. 1b. Boxplots and round dots next to distribution kernels in panels **d** and **e** are defined as in Fig. 1c. Source data are provided with this paper. MPH methylphenidate, SUL sulpiride, arb. units arbitrary units, $k_i^{cer}$ dopamine synthesis capacity index.

dopamine synthesis capacity in the putamen and nucleus accumbens revealed significant effects of methylphenidate (but not sulpiride) on valence-specific reversal signal in the anterior prefrontal cortex (aPFC) and dorsolateral prefrontal cortex (dlPFC; Brodmann area 10 and 46, respectively[49]; Fig. 4a). The effect was strongest for dopamine synthesis capacity in the putamen (Supplementary Results and Supplementary Fig. 7a).

In contrast to the effects in the striatum and visual cortex, methylphenidate's effect in the prefrontal cortex was valence-specific and co-varied positively with dopamine synthesis capacity: methylphenidate increased the prefrontal response to unexpected reward versus punishment signal to a greater degree in participants with higher dopamine synthesis capacity (Fig. 4b and Supplementary Fig. 7b). This effect was driven by reward reversal signals and was not present for punishment reversal signals (Supplementary Fig. 7c). Under placebo, there was a negative association between reward versus punishment reversal signals and dopamine synthesis capacity in the nucleus accumbens (Supplementary Fig. 8).

## Striatal dopamine predicts drug effects on reversal learning
The effects of the dopaminergic drugs on task accuracy were not uniform across the various trial types or individuals, but were significantly qualified by outcome valence, outcome expectancy and striatal dopamine synthesis capacity. Both methylphenidate and sulpiride increased accuracy after unexpected rewards versus unexpected punishments to a greater degree in participants with higher dopamine synthesis capacity (significant in the caudate nucleus for sulpiride; Fig. 5a, c; Supplementary Fig. 10; sulpiride × expectancy × valence × caudate nucleus $k_i^{cer}$: B = 0.070, CI = [0.019, 0.122]; methylphenidate × expectancy × valence × putamen $k_i^{cer}$: B = 0.065, CI = [0.005, 0.125]). Methylphenidate's effect was not significant with synthesis capacity in the caudate nucleus or nucleus accumbens, and sulpiride's effect was not significant with putamen or nucleus accumbens synthesis capacity. The significant effects are visualized in Fig. 5b, d, using voxel-wise PET analyses with the behavioral drug effect as an individual difference predictor.

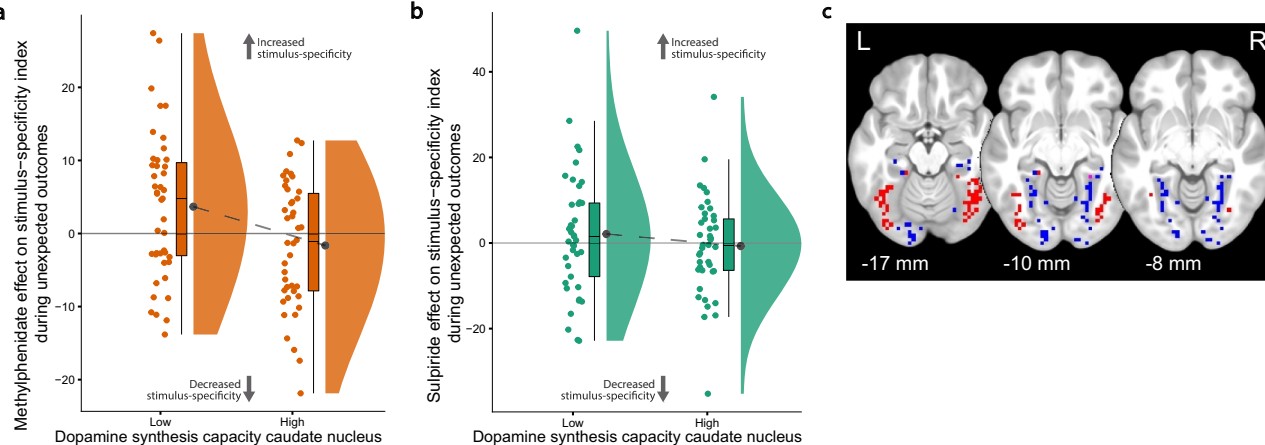

**Fig. 3 | Effects of methylphenidate and sulpiride (relative to placebo) on stimulus specificity of BOLD signal in the fusiform face area (FFA) and parahippocampal place area (PPA) during unexpected outcomes.** Effect of methylphenidate (**a**; $N = 85$ participants) and sulpiride (**b**; $N = 82$) on stimulus-specific reversal-related signal in FFA/PPA as a function of low and high caudate nucleus $k_i^{cer}$ (median split for visualization; methylphenidate versus placebo × expectancy × caudate nucleus $k_i^{cer}$ in repeated-measures ANOVA: $F_{(1,83)} = 7.09$, $P = 0.009$; sulpiride versus placebo × expectancy × caudate nucleus $k_i^{cer}$: $F_{(1,80)} = 5.10$, $P = 0.027$). Boxplots and round dots next to distribution kernels are defined as in Fig. 1c. Source data are provided with this paper. **c** Individually defined ROIs for the FFA (red) and PPA (blue).

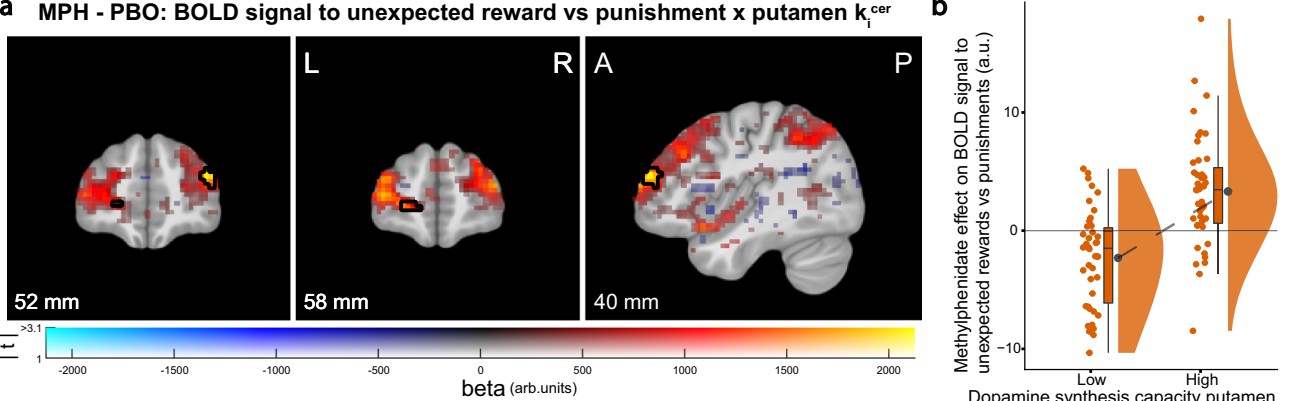

**Fig. 4 | Dopamine synthesis capacity in the putamen predicts methylphenidate effect (relative to placebo) on the prefrontal BOLD signal during unexpected reward versus punishment outcomes. a** Result of univariate regression analysis showing a linear relationship between dopamine synthesis capacity in the putamen and the effect of methylphenidate on BOLD signal related to unexpected rewards minus unexpected punishments (MPH − PBO: (unexpected reward − expected reward) − (unexpected punishment − expected punishment); peak voxel: *x, y,* $z = -20, 59, -2, Z = 5.25$, $p_{cluster\ FWE\ WB} = 0.006$). Figure conventions are as in Fig. 1b. **b** Average contrast estimates extracted from the significant clusters in panel **a** displayed as a function of low and high putamen $k_i^{cer}$ (median split for visualization). Boxplots and round dots next to distribution kernels are defined as in Fig. 1c. $N = 85$ participants in both panels. Source data are provided with this paper. MPH methylphenidate, arb. units arbitrary units, $k_i^{cer}$ dopamine synthesis capacity index.

Inspection of the data and supplementary follow-up analyses that broke down these interactions into their constituent simple interaction effects demonstrated that the methylphenidate effect was primarily due to boosting of punishment reversal accuracy in participants with low dopamine synthesis capacity (Supplementary Results). Conversely, sulpiride boosted reward reversal accuracy, primarily in participants with higher dopamine synthesis capacity. Under placebo, reward versus punishment reversal accuracy was lower for participants with higher caudate nucleus dopamine synthesis capacity (Supplementary Fig. 11; expectancy × valence × caudate nucleus $k_i^{cer}$: B = −0.086, CI = [−0.166, −0.001]).

The pattern of drug effects on RTs generally paralleled those on accuracy. Both methylphenidate and sulpiride decreased RTs for all reward versus punishment predictions to a greater degree in participants with higher dopamine synthesis capacity in the nucleus accumbens and putamen, but not caudate nucleus (Fig. 6a, b; Supplementary Fig. 12; methylphenidate × valence × nucleus accumbens $k_i$ on RTs: B = −0.004, CI = [−0.008, −0.001]; Fig. 6c, d; sulpiride × valence × putamen $k_i$ on RTs: B = −0.005, CI = [−0.009, −0.002]; sulpiride × valence × nucleus accumbens $k_i$ on RTs: B = −0.006, CI = [−0.010, −0.003]). Thus, the drugs induced speeding of punishment predictions compared to reward predictions to a greater degree in participants with lower striatal dopamine synthesis capacity, similar to the improvements in accuracy on update trials.

## Discussion

The present pharmacological fMRI/PET study provides strong support for two pervasive hypotheses about methylphenidate. First, its effects on reversal learning reflect changes in striatal dopamine-related selective output gating of task-relevant cortical signals. Second, interindividual differences in striatal dopamine synthesis capacity explain variability in its cognitive effects, thus strongly establishing the baseline-dependency principle for methylphenidate, which is the most

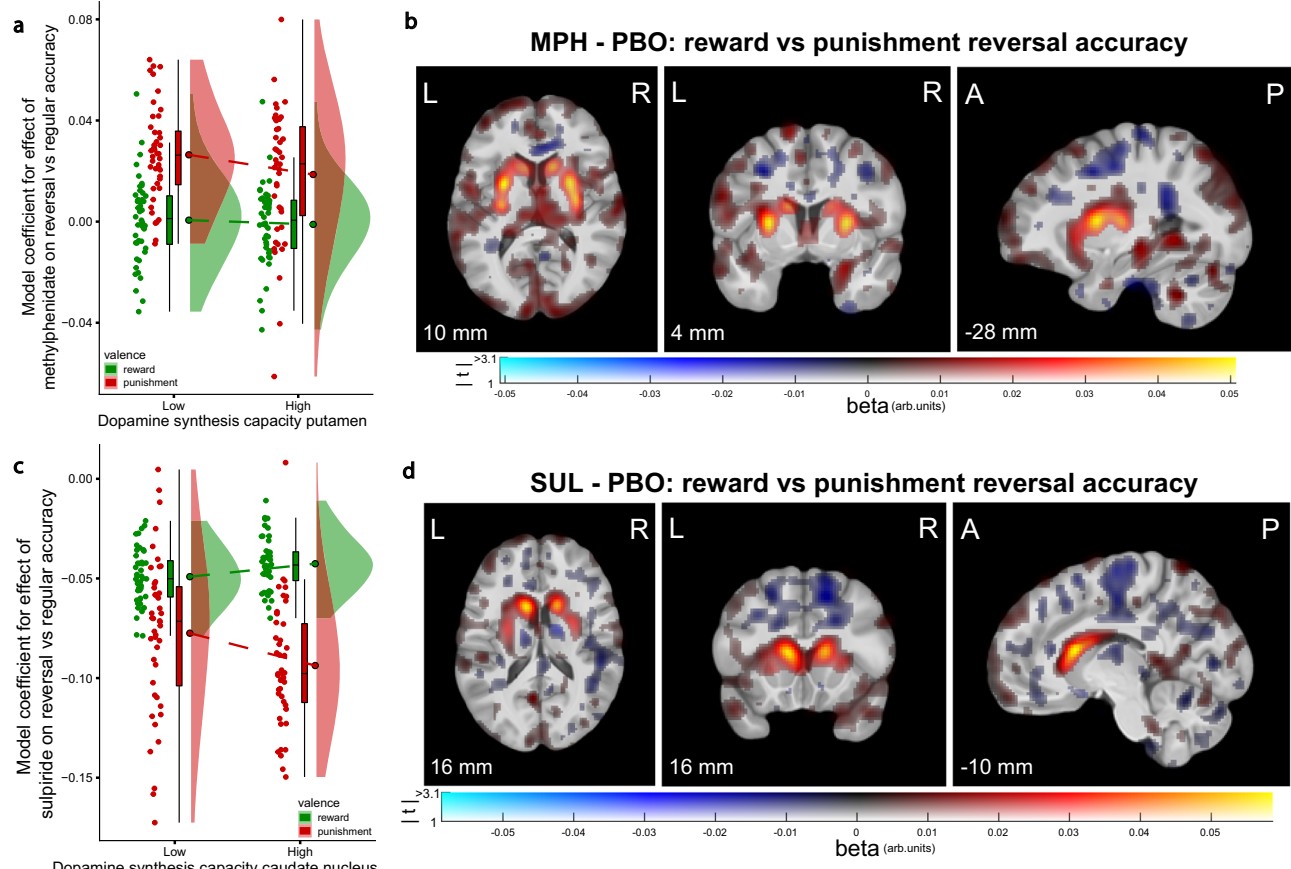

**Fig. 5 | Relationship between dopamine synthesis capacity and drug effects (relative to placebo) on reward versus punishment reversal-learning performance. a** Model coefficients for the effect of methylphenidate on reward and punishment reversal accuracy for participants with low and high dopamine synthesis capacity (median split for visualization). **b** Methylphenidate increased reward versus punishment reversal accuracy to a greater degree in participants with higher striatal dopamine synthesis capacity. Voxel-wise covariation analysis of

the PET $k_i^{cer}$ data with individual participants' model coefficients for the methylphenidate × expectancy × valence interaction effect on accuracy as covariates. **c, d** Same as (**a, b**), but for the effect of sulpiride. Boxplots and round dots next to distribution kernels are defined as in Fig. 1c. Figure conventions of the voxel-wise images are as in Fig. 1b. $N = 88$ participants. Source data are provided with this paper. MPH methylphenidate, SUL sulpiride, arb. units arbitrary units, $k_i^{cer}$ dopamine synthesis capacity index.

prescribed dopaminergic drug together with the related amphetamines (https://clincalc.com/DrugStats/).

While methylphenidate acts on both the dopamine and noradrenaline transporters, a role of noradrenaline in methylphenidate's effects on striatal BOLD signaling is unlikely, given that the noradrenaline transporter is almost absent in the striatum[50]. The role of striatal dopamine was further substantiated by the finding that effects of methylphenidate were paralleled by effects of sulpiride, which acts selectively on the dopamine $D_{2/3}$-receptors that are particularly abundant in the striatum[51]. Both drug effects on striatal BOLD signaling were greater in participants with lower striatal dopamine synthesis capacity.

A striatal locus of action of methylphenidate's effects on the current learning task generally concurs with previous findings showing changes in striatal BOLD signaling during reversal-learning errors after administration of methylphenidate[3] and other dopaminergic drugs[31,52,53]. Furthermore, a striatal locus of action is consistent with work in nonhuman primates showing that methylphenidate's cognitive enhancing effects on choice are related to increased striatal dopamine, measured with microdialysis[54], and [11C]-PE2I PET[55].

The first key contribution of this study is to provide support for the output gating hypothesis of striatal dopamine, according to which dopamine enhances task-relevant cortical representations in proportion to outcome surprise by acting on the striatum[35]. Both methylphenidate and sulpiride modulated BOLD signaling in the FFA to a

greater degree than BOLD signaling in the PPA when participants were presented an unexpected outcome that was paired with a face, and vice versa when they were presented an unexpected outcome paired with a scene. Critically, in contrast to our hypothesis that this output gating would vary with the valence of the unexpected outcome, we did not observe any reward-enhancement of stimulus-selective signals in sensory cortex. Instead, the effects reflect the output of modulating nonspecific surprise or salience signals in the striatum[56], signals that have also been recorded from midbrain dopamine neurons[57,58]. We hypothesize that the reinstatement of stimulus-specific activity in the FFA/PPA upon the presentation of an unexpected outcome reflects an unsigned surprise-induced increase in attentional reorienting to the predictive stimulus in the mind's eye, so that the relevant prediction can be updated. Thus, visual attention to task-relevant information is gated dynamically by outcome surprise signals, conveyed by striatal dopamine. This finding provides a generalization to humans of recent work with nonhuman primates, which demonstrated that dopaminergic prediction error signals modulate cue-selective activity in the visual cortex[59], and that electrically stimulating dopamine neurons in the ventral tegmental area is sufficient for category-selective learning and accompanying fMRI signal changes in the relevant visual, cortical and subcortical areas[60]. Our finding is also in line with previous human work showing that the basal ganglia modulate connectivity with task-relevant posterior sensory areas during salience-driven attention switching[38,39]. While the current experiment cannot demonstrate a

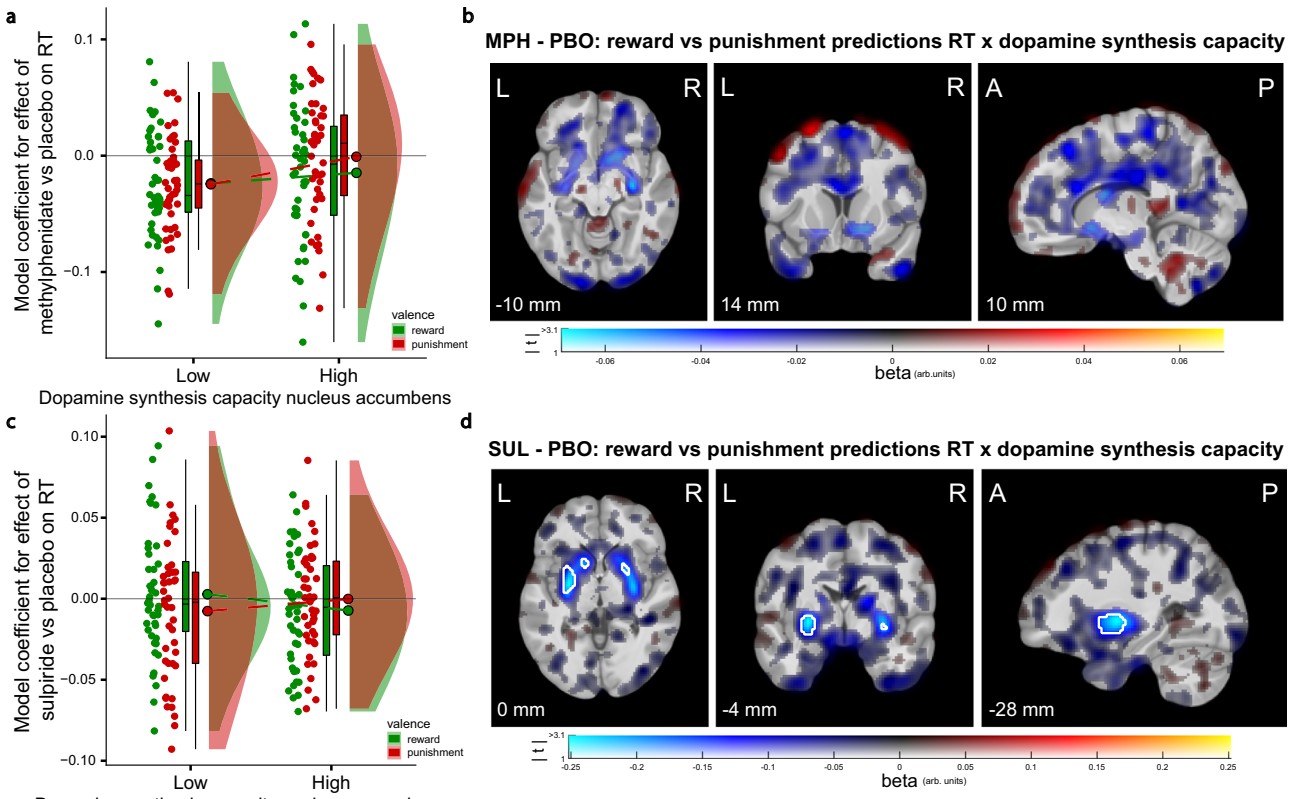

**Fig. 6 | Relationship between striatal dopamine synthesis capacity and drug effects (relative to placebo) on reward versus punishment prediction response times. a** Model coefficients for the effect of methylphenidate versus placebo on response times for only reward and only punishment predictions as a function of dopamine synthesis capacity in the nucleus accumbens (median split for visualization). **b** Voxel-wise covariation analysis of the PET $k_i^{cer}$ data with individual participants' model coefficients for the methylphenidate × valence interaction effect on response times as covariates. **c, d** Same as **a, b**, but for the effect of sulpiride. Boxplots and round dots next to distribution kernels are defined as in Fig. 1c. Figure conventions of the voxel-wise images are as in Fig. 1b. N = 88 participants. Source data are provided with this paper. MPH methylphenidate, SUL sulpiride, PBO placebo, RT response time, arb. units arbitrary units, $k_i^{cer}$ dopamine synthesis capacity index.

causal role for striatal dopamine in gating, a key role for striatal dopamine in this output gating was suggested by the dependency of these visual cortical effects on caudate dopamine synthesis capacity, and by the finding that the effects of methylphenidate were paralleled by those of sulpiride (Fig. 3).

The second key contribution of the present study is that it firmly establishes the baseline dopamine dependency of the effects of methylphenidate and sulpiride on reward-based reversal learning, using a direct measure of striatal dopamine. Both methylphenidate and sulpiride boosted reward relative to punishment reversal accuracy to a greater degree in participants with higher dopamine synthesis capacity. This preregistered pattern of effects resembles those seen in previous pharmacological studies with this paradigm, in which methylphenidate and sulpiride boosted reward versus punishment reversal accuracy to a greater degree in participants with higher baseline working memory capacity, commonly used as an indirect proxy of dopamine synthesis capacity[16,61]. Here we go beyond this prior work by employing a direct measure of baseline dopamine. The pattern of effects from the current large-sample study also mirrors our previous finding from a small sample PET study (N = 11) that the dopamine D$_2$-receptor agonist bromocriptine boosted reward versus punishment reversal accuracy to a greater degree in participants with lower dopamine synthesis capacity[46].

The dopamine synthesis-dependent effect of methylphenidate on behavior was accompanied by greater increases in reward versus punishment-related reversal signals in the lateral aPFC (Fig. 7). While this finding concurs generally with prior results showing modulation by methylphenidate of reward-related neural activity in the prefrontal cortex during reversal learning[3], our results demonstrate that these

effects depend on striatal dopamine synthesis capacity. The striatal dopamine dependency of valence-specific modulation of learning is predicted by models of striatal dopamine's role in reinforcement learning[62] and suggests that methylphenidate acts on valence-dependent release of striatal dopamine to modulate valence-dependent signaling to the aPFC, consistent with classic animal lesion studies implicating both striatum and aPFC in reversal learning, specifically the orbitofrontal cortex (OFC)[42,63]. The specific locus of the present effect also concurs remarkably well with an extensive body of work implicating the lateral OFC, and its interactions with striatal dopamine signaling, in acquiring and reversal learning of stimulus-outcome associations[64–71].

What mechanisms account for this pattern of baseline synthesis-dependent effects on behavior? In the past, we have argued, based on popular Go-NoGo models of dopamine in the basal ganglia[34,72], that dopamine D$_2$-receptor agonists likely act postsynaptically in "low-dopamine participants" with sensitized postsynaptic D$_2$-receptors, while acting presynaptically in participants with higher dopamine synthesis capacity and putatively sensitized autoreceptors[46]. The effects of the D$_2$-receptor antagonist sulpiride observed here might also reflect paradoxical presynaptic effects, particularly in participants with high synthesis capacity, which leads to a net increase in striatal dopamine release, biasing the system towards better reward learning[34,40,41]. A presynaptic action of sulpiride in participants with high baseline dopamine levels is substantiated by the finding that sulpiride speeded responses for reward predictions compared with punishment predictions to a greater degree in participants with higher dopamine synthesis capacity (Fig. 6c, d). While the relatively low dose of sulpiride used here is also consistent with presynaptic action[34,73], we

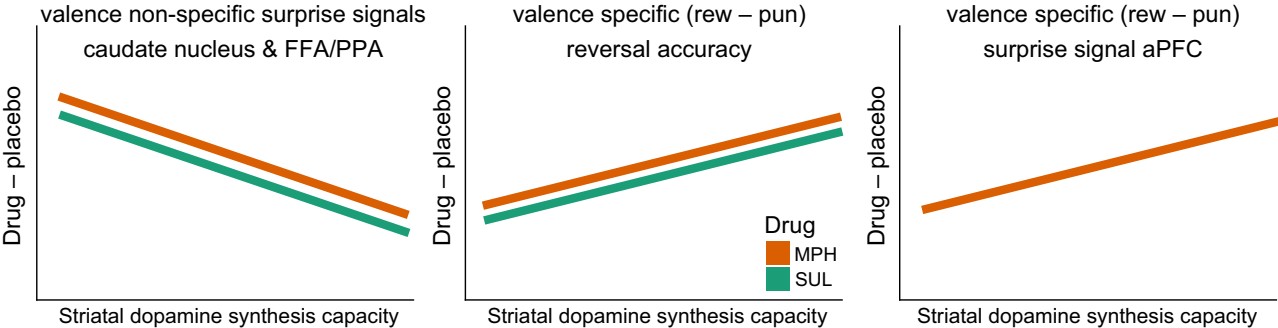

**Fig. 7 | Schematic of key findings.** The dopaminergic drugs methylphenidate and sulpiride boosted outcome surprise signals in the caudate nucleus and stimulus-specific visual association cortex in 'low striatal dopamine' participants (left panel). This was accompanied by the drugs boosting punishment compared with reward-based reversal learning in the "low striatal dopamine" participants, while boosting relative reward-based reversal learning to a greater degree in "high striatal dopamine" participants (middle panel). Methylphenidate also boosted the associated prefrontal surprise signal specifically for reward versus punishment to a greater degree in "high striatal dopamine" participants (right panel). MPH methylphenidate, SUL sulpiride, FFA fusiform face area, PPA parahippocampal place area, aPFC anterior prefrontal cortex, rew reward, pun punishment.

cannot exclude the possibility that it acted also postsynaptically, perhaps in line with the overall reduction in accuracy across participants[74].

The valence-specific effects of methylphenidate resembled those of sulpiride and might reflect increases in tonic striatal dopamine levels[19]. One speculative possibility is that methylphenidate differentially disturbed the balance between temporally precise dopamine release and slow reuptake, depending on dopamine synthesis capacity. This hypothesis builds on early proposals that methylphenidate alters impulse-dependent dopamine release by eliciting inhibitory feedback from raised tonic levels of dopamine acting at dopamine $D_2$-autoreceptors[26,75–77]. We speculate that the greater dynamic range for boosting dopamine tone by methylphenidate in participants with lower dopamine synthesis capacity might be accompanied by reduced dynamic range for reward-elicited dopamine release. Concomitantly, the dynamic range for punishment-elicited dopamine dips might be enhanced due to the elevated reference point against which negative prediction errors are compared. We hypothesize that methylphenidate altered the balance between a phasic dopamine mode that promotes reward/punishment-specific learning and a tonic dopamine mode that promotes surprise-driven attention, depending on baseline dopamine synthesis capacity. While speculative, this hypothesis is consistent with the finding that the greatest relative decreases in reward versus punishment-related reversal accuracy were seen in those participants (i.e., "low-dopamine participants") who also exhibited the largest increases in indices that likely reflect changes in arousal and attention, i.e., valence-nonspecific speeding and accuracy (Figs. 5a and 6a), task-nonspecific BOLD signals in the bilateral parietal cortex (Supplementary Fig. 9c), and valence-nonspecific striatal and FFA/PPA BOLD signal (Figs. 2, 3, and 7).

The finding that sulpiride's effects on reward learning varied positively with striatal dopamine synthesis capacity in healthy participants is remarkably reminiscent of findings that antipsychotic therapy response is greater in psychotic patients with higher dopamine synthesis capacity[78–80]. This is pertinent given the implication of reward-based reversal-learning deficits in schizophrenia[81]. Conversely, the finding that methylphenidate boosts unsigned surprise-related striatal signals and stimulus-selectivity in the visual cortex, while also reducing the impact of reward versus punishment signals, to a greater degree in participants with lower dopamine synthesis capacity, has relevance for ADHD, which has been associated with low striatal dopamine synthesis capacity[82]. The results raise the question whether those individuals who are most sensitive to methylphenidate's attention-enhancing potential are also best protected against its potential for compulsive reward-associated behaviors[83].

This study raises a number of open questions. First, why does sulpiride boost nonspecific surprise signals in the caudate nucleus and stimulus-specific visual cortex to a greater degree in participants with lower dopamine synthesis capacity? One possibility is that this striatal BOLD signal increase reflects a tonic blockade of $D_2$-mediated suppression of indirect (NoGo) pathway activity, and a concomitant increase in the salience of unsigned surprise signals, thus promoting attention shifting after unsigned surprise. A second question is how to reconcile the present finding that methylphenidate and sulpiride-related boosting of reward-specific learning was greater in participants with higher dopamine synthesis capacity with our previous finding (in a subset of the same participants) that drug-related boosting of effort-based value and choice was greater in participants with lower dopamine synthesis capacity[9]? In line with prior work and the proposal put forward above, we argue that this might reflect differential roles of phasic and tonic dopamine in learning from reward and in the expression of learned value on choice[84]. Third, the negative effect of interindividual differences in striatal dopamine synthesis capacity on reward versus punishment learning under placebo stands in contrast to a positive effect of striatal dopamine synthesis capacity on this same task in a previous study[46]. This likely reflects the use of [$^{18}$F]DOPA as opposed to [$^{18}$F]FMT for estimating dopamine synthesis capacity. Both ligands are substrates for aromatic amino acid decarboxylase, but [$^{18}$F]DOPA is subject to additional in vivo metabolism, implying lower signal-to-noise ratio than [$^{18}$F]FMT, and perhaps also to some degree reflecting dopamine turnover rather than synthesis capacity[85]. Strikingly, there is precedent in the literature for opposite effects of [$^{18}$F]FMT and [$^{18}$F]DOPA, for example, as a function of aging[85–87], as well as $D_2$-receptor binding potential[88,89]. While the source of these contrasting effects remains to be elucidated, one might ask whether this relates to differential contributions of [$^{18}$F]DOPA and [$^{18}$F]FMT to tonic dopamine levels and neuronal integrity, respectively[90,91]. A final puzzle is why we did not observe effects of sulpiride on reward-specific BOLD signal (in the aPFC or elsewhere), despite sulpiride-related changes in reward-specific reversal learning. One possibility is that sulpiride's effects on phasic dopamine signals were not sufficiently prolonged to be detectable in terms of BOLD signal changes.

It is important to note that the current results reflect effects of acute administrations of fixed doses of methylphenidate and sulpiride, whereas these drugs are typically taken repeatedly as long-term treatment of ADHD and psychosis in a variety of doses. Thus, the current results do not address potential long-term effects on brain and behavior, and higher doses might produce different effects, particularly for sulpiride, which elicits stronger postsynaptic effects at higher doses[73]. In addition, the current design with fixed timings between

drug intake and task performance, and without monitoring of drug plasma levels, did not enable us to control for interindividual differences in drug plasma concentrations during scanning. A second consideration is the fact that sulpiride also acts as an antagonist of 5-HT1A receptors. However, we consider it less plausible that the effects observed here reflect changes in serotonin receptor activity, because such effects would be more substantial only at higher doses of sulpiride, due to its low affinity for the 5-HT1A receptor[92]. Finally, another important issue to consider in all pharmacological fMRI studies is the degree to which BOLD signal changes reflect non-neuronal drug effects, such as changes in cerebral blood flow or cerebrovascular reactivity in the absence of changes in neural activity[93]. We consider this unlikely in this case because the effects were regionally selective and task-specific. Specifically, the effect of methylphenidate on striatal BOLD signal was observed during unexpected relative to expected outcomes. Moreover, it was observed only in the striatum, and did not extend to other regions activated by the unexpected outcomes (Fig. 1b). Similarly, the effect of methylphenidate on aPFC BOLD signal was observed during unexpected reward relative to punishment outcomes.

Collectively, the finding that both methylphenidate and sulpiride's effects on reversal learning and associated brain signals depend on striatal dopamine synthesis capacity firmly establishes the baseline dopamine dependency hypothesis. It extends prior evidence from smaller-scale PET studies demonstrating baseline dopamine-dependent effects of dopaminergic drugs, including psychostimulants, on cognitive tasks in humans[2,7,9,46,47], as well as from studies using proxy measures of dopamine function, such as working memory or trait impulsivity[16,61,94,95]. The finding also concurs with accumulating evidence from clinical studies that dopamine synthesis capacity predicts the effectiveness of antipsychotics[78–80]. The large interindividual variability in methylphenidate's effects highlights that caution is warranted when using psychostimulants for cognitive enhancement[2,9,46–48], and opens avenues for assessing [$^{18}$F]DOPA PET imaging as a biomarker for identifying those for whom methylphenidate will confer the greatest benefits and the smallest risks.

## Methods
### Participants
One hundred healthy volunteers, 50 women and 50 men, were recruited for the study (age at inclusion: range 18–43, mean (SD) = 23.0 (5.0) years). All participants provided written informed consent and were paid 309 euro on completion of the study. The study was approved by the local ethics committee ("Commissie Mensgebonden Onderzoek", CMO region Arnhem-Nijmegen, The Netherlands: protocol NL57538.091.16). People were recruited via an advertisement on the Radboud University electronic database for research participants, and via advertisement flyers around Nijmegen. Prerequisites for participation were an age between 18 and 45 years, Dutch as native language and right-handedness. Before admission to the study, participants were extensively screened for adverse medical and psychiatric conditions. Exclusion criteria included any current or previous psychiatric or neurological disorders, having a first-degree family member with a current or previous psychiatric disorder, clinically significant hepatic, cardiac, renal, metabolic or pulmonary disease, epilepsy, hyper or hypotension, habitual smoking or drug use, pregnancy, and MRI contraindications, such as unremovable metal parts in the body and claustrophobia (more details in ref. 96).

Six participants dropped out because of discomfort in the MRI or PET scanner ($N = 4$), personal reasons ($N = 1$) or technical failure of the PET scanner ($N = 1$). Six more failed to meet the task performance criterion (described below) on one of the sessions ($N = 6$). For the fMRI analyses, three additional participants were excluded from contrasts comparing methylphenidate and placebo, and six were excluded from contrasts comparing sulpiride and placebo, because of poor data

quality on one of the sessions (Supplementary Methods). This resulted in $N = 88$ for the behavioral analyses, and $N = 85$ and $N = 82$ for the effects of methylphenidate and sulpiride, respectively, in the fMRI analyses.

### General procedure and pharmacological manipulation
Data were collected as part of a large PET, pharmaco-fMRI study on the effects of methylphenidate and sulpiride on brain and cognition, employing a within-subject, placebo-controlled, double-blind crossover design. The number of participants in the current work was determined by the number of participants included in the overarching project, rather than with an a priori power calculation for the specific task reported here. Not all the outcome measures of the registered trial for the overarching project are reported in this paper. For a detailed description of the testing sessions and tasks and measures collected in the overarching project, see ref. 96.

The study consisted of five testing days separated by at least one week. The first was an intake session in which participants were screened for inclusion criteria, an anatomical MRI scan was obtained, several baseline measures were collected, and the reversal-learning task was explained and briefly practiced.

The second, third and fourth testing days were six-hour-long pharmaco-fMRI sessions in which participants performed the reversal-learning task and a battery of other tasks not reported here (for two of them see refs. 9,47). The three sessions were identical except for the pharmacological manipulation. Participants received an oral administration of 20 mg of the monoamine transporter blocker methylphenidate on one day, 400 mg of the dopamine $D_{2/3}$-receptor antagonist sulpiride on another, and a placebo on a third day. The order was randomized by an independent researcher and the medication was prepared and coded by the pharmacy in accordance with the pre-specified randomized order, ensuring that the experimenters, as well as the participants, were blind to the drug status on each session. Each morning, the participant briefly practiced the reversal-learning task off drug, and in the afternoon performed the task on drug in the MRI scanner. Drug administration timings were optimized to have peak drug effects during this fMRI paradigm. The reversal-learning task started -1.5 h after methylphenidate administration and 3 h after sulpiride administration. The mean time to maximal plasma concentration of methylphenidate is -1.5 h with a half-life of approximately 6 h[97]. For sulpiride, this is approximately 3 h with a half-life of -12 h[98]. To account for the difference in peak times of methylphenidate and sulpiride and not break blindness to drug status, we used a double-dummy design. Participants received (supervised) two identical capsules on each day, one 3 h before the task and one 1.5 h before the task. One of the capsules was a placebo and the other contained the drug (or another placebo on the placebo session). The dose selection was based on previous studies which had revealed significant effects and good tolerance[3,16,61,99]. Blood pressure, heart rate, medical symptoms, and mood measures were monitored three times each session: before the start of the task battery, 20 min after intake of the second capsule, and after the task battery (results reported in ref. 96).

On the fifth day, participants underwent an [$^{18}$F]DOPA PET scan of the brain to measure their baseline dopamine synthesis capacity. Fifty minutes before the PET scan started, participants received 150 mg of carbidopa and 400 mg of entacapone to minimize peripheral metabolism of [$^{18}$F]DOPA by peripheral decarboxylase and catechol-O-methyltransferase (COMT), respectively, thereby increasing the signal-to-noise ratio in the brain[100–103].

### Reversal-learning paradigm
The reversal-learning paradigm (Fig. 1a) was the same as used previously[61]. It was programmed in Presentation (Neurobehavioral Systems; version 18.3 03.11.16) and run under Windows 7 Enterprise OS, on a DELL PRECISION T3500 computer. The task was performed in

the MRI scanner, where visual stimuli were presented on a screen visible to the participant via a mirror mounted on the head coil. Responses were collected using an MR-compatible response pad (Current Designs, Inc; Philadelphia, PA, USA).

In each trial two black-and-white images, one of a neutral face and the other of a landscape scene, were presented vertically adjacent on the screen (location randomized) on a gray background. One of the images was associated with a reward outcome and the other with a punishment outcome, with 100% certainty. Unlike typical probabilistic reversal-learning tasks, participants did not choose between the two stimuli. Instead, one stimulus was randomly selected by the computer, indicated by a black border highlighting the image, and the participants were asked to predict whether the selected stimulus was associated with a reward or a punishment. After the prediction was made, the actual outcome associated with the selected stimulus was presented. Hence, the reward and punishment feedback was not instrumental but of Pavlovian nature, because it did not depend on the participant's choice but was directly associated with the stimulus. With this prediction design, both the frequency of occurrence and the need for behavioral adjustment are matched between reward and punishment reversal trial types, unlike more typical instrumental reversal-learning tasks, which require choosing the rewarded stimuli and where reversals are typically signaled by punishment prediction errors.

After the presentation of the two stimuli, there was a 1500 ms response window, during which participants predicted reward or punishment outcomes by pressing a button with the right index or middle finger (counterbalanced across participants). After a response was made, there was a 1000 ms delay before the outcome was presented for 500 ms in the middle of the screen between the two image stimuli. Then the intertrial interval followed for 1000–3500 ms, during which only a fixation cross was presented. The interval of each trial was drawn from a Poisson distribution with a mean at 2500 ms. A reward outcome consisted of a green smiley and a + €100 sign. A punishment consisted of a red sad smiley and a -€100 sign. If there was no response within the response window, a "Too late" message was displayed instead of the outcome. Participants were not compensated for the monetary winnings, because they did not depend on performance and the number of reward and punishment outcomes were equated.

Our primary outcome measure was the accuracy on reversal trials. The reversal of the stimulus-outcome contingencies was signaled by the unexpected presentation of a reward outcome for the stimulus that was previously associated with punishment (or vice versa). The first trial after such an unexpected outcome was marked as a reversal trial. The accuracy on these trials reflects how well participants were able to update their stimulus-outcome associations after unexpected rewards or punishments. On reversal trials, the same stimulus was highlighted as on the previous trial with the unexpected outcome to match motor switching and prediction updating requirements between valence conditions. Reversals happened after reaching a learning criterion of 3–6 consecutive correct predictions, drawn from a Poisson distribution that made criteria 4 and 5 more likely than 3 and 6. This variability in the criterion prevented the predictability of reversals. If the participant made the incorrect prediction on a trial where an unexpected outcome was going to occur, the contingencies did not reverse, the trial was simply marked incorrect and the count to learning criterion was reset.

Participants performed 3 blocks of 119 trials in the scanner, resulting in a total of 357 trials per session (-35 min). The number of reversals that occurred is an indication of overall task performance because reversals only happened after several consecutive correct predictions. We considered fewer than 20 reversals on a session a sign of task disengagement and excluded participants with fewer than 20 reversals on one of the sessions from the analyses ($N = 6$; Supplementary Fig. 15). The average number of reversals per session was 46 (SD = 9), 23 for reward, and 23 for punishment.

## MRI acquisition and preprocessing

The MRI experiment was performed on a 3 T Siemens Magnetom Skyra MRI scanner at the Donders Institute, using a 32-channel head coil. Images with blood-oxygen-level-dependent (BOLD) contrast were acquired in three runs, using a whole-brain T2*-weighted gradient echo multi-echo echo planar imaging (EPI) sequence (38 slices per volume; interleaved slice acquisition; repetition time, 2320 ms; echo times, 9 ms, 19.3 ms, 30 ms, and 40 ms; field of view: $211 \times 211$ mm; flip angle 90°; $64 \times 64$ matrix; 3.3 mm in-plane resolution; 2.5 mm slice thickness, 0.4 mm slice gap). On the intake session a whole-brain structural image was acquired for within-subject registration purposes, using a T1-weighted magnetization prepared, rapid-acquisition gradient echo sequence (192 sagittal slices; repetition time, 2300 ms; echo time, 3.03 ms; field of view: $256 \times 256$ mm; flip angle, 8°; $256 \times 256$ matrix; 1.0 mm in-plane resolution; 1.0-mm slice thickness).

All MRI data were preprocessed using fMRIPrep (1.2.6-1; RRID:SCR_016216[104,105]). Before preprocessing the functional data with fMRIPrep, we combined the multi-echo data into a single time series per fMRI run with the multi-echo toolbox (https://github.com/Donders-Institute/multiecho; commit Nr.: 9356bc51ef) using the TE algorithm, in which the different echoes are weighted by their echo time. The multi-echo-combined functional scans were then realigned, coregistered to the participant's T1-weighted anatomical scan, and spatially normalized to MNI152 space with fMRIPrep. More details on the fMRI preprocessing with fMRIPrep can be found in the Supplementary Methods. We used Statistical Parametric Mapping 12 (SPM12; https://www.fil.ion.ucl.ac.uk/spm/software/spm12/) running in MATLAB R2018b (Mathworks Inc.; https://nl.mathworks.com/products/matlab.html) to spatially smooth the final preprocessed BOLD time series with a 6 mm FWHM kernel.

## PET acquisition and preprocessing

The brain PET data were acquired on a state-of-the-art PET/CT scanner (Siemens Biograph mCT; Siemens Medical Systems, Erlangen, Germany) at the Department of Medical Imaging of the Radboud University Medical Center. We used the well-validated radio-tracer [18F]DOPA, which was synthesized at Radboud Translational Medicine BV (RTM BV) in Nijmegen. The tracer is a substrate for aromatic amino acid decarboxylase, the enzyme that converts DOPA into dopamine. The rate of conversion of [18F]DOPA into dopamine provides an estimate of dopamine synthesis capacity.

The procedure started with a low-dose CT scan to use for attenuation correction of the PET images. Then, the [18F]DOPA tracer was administered (-185 MBq) via a bolus injection in the antecubital vein and the PET scan was started. Dynamic PET data ($4 \times 4 \times 3$-mm voxel size; 5-mm slice thickness; $200 \times 200 \times 75$ matrix) were acquired over 89 min and divided into 24 frames ($4 \times 1$, $3 \times 2$, $3 \times 3$, $14 \times 5$ min). Data were reconstructed with weighted attenuation correction and time-of-flight recovery, scatter corrected, and smoothed with a 3-mm FWHM kernel.

The PET data were preprocessed and analyzed using SPM12. All frames were realigned to the mean image to correct for head motion between scans. The realigned frames were then coregistered to the structural MRI scan, using the mean PET image of the first 11 frames (corresponding to the first 24 min), which has a better range in image contrast outside the striatum than a mean image over the whole scan time. Presynaptic dopamine synthesis capacity was quantified as the tracer influx rate $k_i^{cer}$ (min$^{-1}$) per voxel with graphical analysis for irreversible tracer binding using Gjedde-Patlak modeling[106,107]. The analysis was performed on the images corresponding to 24–89 min, which is the period after the irreversible compartments had reached equilibrium and the input function to the striatum had become linear. The $k_i^{cer}$ values represent the rate of tracer accumulation relative to the reference region of cerebellar gray matter, where the density of dopamine receptors and metabolites is extremely low compared with

the striatum[50,108]. The cerebellar gray matter mask was obtained using FreeSurfer segmentation of each individual's anatomical MRI scan, as implemented in fMRIPREP. Resulting $k_i^{cer}$ maps were spatially normalized to MNI space, smoothed with an 8-mm FWHM kernel, and brain extracted.

After preprocessing, we extracted the mean $k_i^{cer}$ values from the caudate nucleus, putamen, and nucleus accumbens in native subject space. These extracted $k_i^{cer}$ values were used as a covariate in the behavior and fMRI analyses. See Supplementary Table 2 for the correlations between $k_i^{cer}$ values in the three ROIs. The striatal ROI masks were obtained from an independent, functional connectivity-based parcellation of the striatum conducted in a previous study[109] (Supplementary Fig. 16). That study used a clustering method to identify five striatal subregions based on resting state functional connectivity, which we combined into our three ROI masks. These masks map well onto the anatomical structures of the caudate nucleus, putamen and nucleus accumbens.

### Statistical analysis

**Behavior**. Statistical analyses of the behavioral measures were performed with Bayesian mixed-effects regression models using the brm function of the brms package (version 2.14.0)[110,111] in R (version 4.0.1)[112]. We modeled trial-by-trial accuracy with a Bernoulli response distribution and logit link function, and response times were modeled with a lognormal response distribution. Note that the resulting effect estimates of response times are therefore on the log scale, and those for accuracy are on the logit scale. Exponentiating the logit-scale estimate gives the odds ratio of an effect. We used default noninformative priors, and trials without a response were excluded from the analyses. Model coefficients for the estimated effects were deemed statistically significant if their 95% credible interval did not contain zero.

The dependent variables were modeled as a function of four independent variables: drug (methylphenidate or sulpiride versus placebo), response update type (i.e., reversal trials [update response] versus regular trials [no update]), valence (predict reward versus predict punishment), and dopamine synthesis capacity. The factors were sum-to-zero coded, and the continuous variable for dopamine synthesis capacity was scaled and mean-centered. All factors and all interactions between them were included as both fixed effects and random effects with participants as grouping factor (also including random intercepts), except for the between-subject variable of dopamine synthesis capacity, which was only a fixed effect. To test for drug effects with proper effect coding, we excluded trials from the sulpiride session in the models for effects of methylphenidate versus placebo, and excluded trials from the methylphenidate session in models for effects of sulpiride versus placebo. Significant interaction effects were broken down and tested on the contribution of their constituent simple effects in reduced models. The model coefficients of such reduced models were used as data points for figures displaying interaction effects split up by levels of a specific factor.

We estimated each model separately for dopamine synthesis capacity values extracted from the caudate nucleus, putamen or nucleus accumbens. Our confirmatory analyses focused on dopamine synthesis capacity in the caudate nucleus, and we additionally explored the effect of dopamine synthesis capacity in the putamen and nucleus accumbens. For visualization of the effects of dopamine synthesis capacity, we performed voxel-wise regression analyses of the effects of interest on the PET $k_i^{cer}$ data, using SPM12.

**fMRI**. All fMRI analyses were performed using SPM12. We modeled participants' event-related BOLD responses with one general linear model (GLM) for all three drug sessions. The four different outcome types (expected punishment, expected reward, unexpected punishment and unexpected reward) were modeled separately for outcomes associated with the face stimulus and the scene stimulus, resulting in

eight task parameters. Only trials with correct responses were included in the models (note that an unexpected outcome due to a reversal did not render the response on that trial incorrect). The separated outcome types per face/scene stimulus were used to extract stimulus-specific contrast estimates from the FFA/PPA (see below). The regular contrasts were created from the four outcome types with face and scene-related signal combined.

In addition to the task parameters, we included confound regressors: six realignment parameters, framewise displacement, global CSF and global white matter signal, six anatomical principal component noise regressors (aCompCor), and all independent components labeled as noise by ICA-AROMA (the different number for each fMRI run). We included the ICA-AROMA noise components as regressors in the GLM rather than using AROMA-denoised data, because denoising the data also removes variance that is potentially shared between noise regressors and task parameters. The regressors were created by convolving delta functions at the onset of the presentation of the outcomes with the SPM standard hemodynamic response function. Low-frequency drifts in the data were controlled using a high-pass filter with a 128 s cutoff.

For each drug separately, we created contrasts for the effects of expectancy ([unexpected reward − expected reward] + [unexpected punishment − expected punishment]), valence ([unexpected reward − unexpected punishment] + [expected reward − expected punishment]), and the expectancy × valence interaction ([unexpected reward − expected reward] − [unexpected punishment − expected punishment]). The contrasts for the drug effects were created by subtracting the contrast vectors for placebo from the contrast vectors for methylphenidate/sulpiride. The resulting contrast images of each participant were taken to the group-level analysis and submitted to a one-sample $t$ test, with participants' striatal dopamine synthesis capacity values as covariates in the analysis in three separate models for each striatum ROI. We report striatal activation results for clusters surviving peak-level family-wise error (FWE) correction at $P < 0.05$ after small-volume correction (SVC) for the combination of the three striatal ROI masks (indicated with $p_{peak\ FWE\ SVC}$). We also performed whole-brain analyses, for which we report activations surviving cluster-level FWE correction at $P < 0.05$ (indicated with $p_{cluster\ FWE\ WB}$), with an initial uncorrected cluster-defining threshold of $P < 0.001$. In Supplementary Table 3, we report all activations that were significant at a whole-brain threshold of $P < 0.001$, uncorrected.

For visualization, we have used a dual-coded approach displaying both the $t$ values and beta values[113,114]. The blue-black and red-yellow colors code the beta values. The $t$ values are displayed in terms of the opaqueness of the colored blobs. All blobs that pass the threshold for significance are fully opaque; the rest are more transparent the lower the associated $t$ value and fully transparent below a $t$ value of 1 (to reduce clutter). Visualizing the data in this way is more revealing than only displaying $t$ values, which hides all effects that do not reach the cutoff score for statistical significance.

**Stimulus specificity of FFA/PPA signal**. To assess drug effects on stimulus-specific activity in the relevant visual association cortex, we analyzed a stimulus-specificity index for BOLD signal in the FFA/PPA related to outcomes after faces versus scenes.

First, we created individually defined ROI masks for the FFA and PPA. For each participant, we estimated a separate GLM to model the BOLD response to the face and scene stimulus, using two regressors for the onsets of the stimuli on trials where the face or scene was highlighted, respectively. The same confound regressors as in the other GLMs were included. Based on that GLM, two contrast images were created: BOLD response to faces minus scenes, and for scenes minus faces. The FFA ROI was defined as a sphere with 3-mm radius around the peak voxel of the faces minus scenes contrast within an anatomical mask of the fusiform gyrus. The PPA ROI was a 3-mm

sphere around the peak voxel of the scenes minus faces contrast within the mask of the combination of the parahippocampal and lingual gyri. The anatomical bounding masks were created using the WFU_PickAtlas toolbox for SPM12 (https://www.nitrc.org/projects/wfu_pickatlas/).

Second, we created a stimulus-specificity index per outcome type, using the main GLM with outcome regressors split by their association to the face or scene stimulus. For each outcome type we created a contrast for the elicited BOLD response when presented after the face versus scene stimulus (e.g., [unexpected reward after a face – unexpected reward after a scene]). We then extracted the average contrast estimates in the individually defined FFA (where positive values indicate greater outcome signal after the face compared with scene stimulus) and PPA (where negative values indicate greater outcome signal after the scene compared with face stimulus). The final stimulus-specificity index was created by subtracting estimates in the PPA from those in the FFA (i.e., FFA minus PPA), resulting in a measure with greater values indicating greater stimulus specificity of the outcome-related signal in the relevant ROI.

Finally, the stimulus-specificity indices were analyzed in a repeated-measures ANOVA with drug status, outcome expectancy and outcome valence as within-subject factors, and dopamine synthesis capacity in the caudate nucleus as between-subjects factor. In addition, we explored the effect of dopamine synthesis capacity in the putamen and nucleus accumbens in separate ANOVA models. The ANOVA models were run using the aov_car function from the afex package (version 0.28-0)[115] in R (version 4.0.2)[112].

### Reporting summary

Further information on research design is available in the Nature Research Reporting Summary linked to this article.

## Data availability

The minimally processed data used in this study and the overarching project it is part of are available from the Donders Institute Data Repository (https://doi.org/10.34973/wn51-ej53). The final data derivatives relevant to the current work are available from a separate collection on the Donders Institute Data Repository (https://doi.org/10.34973/bc23-mz79). Source data are provided with this paper.

## Code availability

All code for data processing, analysis, and figure creation is available with an accompanying readme file from the Donders Institute Data Repository (https://doi.org/10.34973/bc23-mz79).

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

## Acknowledgements

We thank Margot van Cauwenberge, Peter Mulder and Monique Timmer for medical assistance during data collection, Marieke van der Schaaf for the task presentation code, and we thank the people that participated in this study. The work was funded by a Vici grant to R.C. from the Netherlands Organization for Scientific Research (NWO; Grant No. 453-14-015). A.W. was funded by an NIH Grant (F32MH115600-01A1). This project has received a Voucher from the European Union's Horizon 2020 Framework Programme for Research and Innovation under the Specific Grant Agreement No. 945539 (Human Brain Project SGA3).

## Author contributions

Conceptualization: R.C. Data curation: R.v.d.B., J.I.M., B.L., D.P., and L.H. Formal analysis: R.v.d.B. and R.C. Software: R.v.d.B. and B.L. Investigation: R.v.d.B., B.L., J.I.M., D.P., L.H., and R.J.V. Writing: R.v.d.B. and R.C. Review and editing: R.v.d.B., B.L., J.I.M., L.H., D.P., A.W., R.J.V., J.B., and R.C. Project administration: J.I.M. and R.C. Supervision: R.C. and J.B. Funding acquisition: R.C.

## Competing interests

The authors declare no competing interests.
