## [Peer Review File · Nature Communications]

Striatal dopamine dissociates methylphenidate effects on value-based versus surprise-based reversal learningREVIEWER COMMENTS

Reviewer #1 (Remarks to the Author):

This human study tests the overall hypothesis that the effects of methylphenidate (MPH) on reversal learning performance depend on baseline levels of striatal dopamine and that these effects are driven by a gating mechanism, modulating neural responses to unexpected outcomes in stimulus-specific sensory cortex. The study combines dopamine pharmacology (catecholamine reuptake inhibitor MPH and D2/3 antagonist sulpiride [SUL]), task-based functional magnetic resonance imaging (fMRI), and positron emission tomography (PET) to measure dopamine (DA) synthesis capacity. The key result is that while MPH increased and SUL decreased behavioral performance, neural responses in the striatum to rewarding and punishing unexpected outcomes are enhanced by both MPH and SUL, particularly in subjects with low DA synthesis capacity. Paralleling these findings, both drugs enhanced responses to unexpected outcomes in stimulus-specific sensory cortex in a DA synthesis capacity-dependent way, providing support for the gating hypothesis of striatal DA. In addition, valence-specific responses to unexpected outcomes in the prefrontal cortex (PFC) were found to be modulated by MPH, particularly in subjects with high DA synthesis capacity in the striatum.

This is an extremely important study, addressing a long-standing and significant question in cognitive neuroscience. There are many strengths to this manuscript. The design is very strong with a large sample size (N=100), measurements of baseline DA synthesis capacity, and two different dopaminergic drugs, administered in a double-blind and placebo-controlled fashion. Event-related fMRI during a well-established reversal learning task is another strength. The manuscript is also well-written. The results will have a substantial impact on the field. I have a number of relatively minor comments suggestions, mostly to improve clarity.

Specific comments:

1. The results section does not always specify what comparisons are used to determine the effects of MPH and SUL. I assume that these are relative to placebo, but the existence of a placebo condition is not even mentioned before line 212. Are the behavioral (lines 132-136, Figure 1D-E) and neural effects (lines 161-165, Figure 2A-E, but also Figure 3A&B, Figure 4 B, and Figure 5B&D) of MPH and SUL computed relative to placebo? Similarly, it should be made clear what is meant by “the drugs' effect” in line 181? This needs to be clarified throughout the manuscript.

2. Why are the effects for SUL (Figure 2E) extracted from the cluster identified in Figure 2C, and not the cluster defined in Figure 2A (as it is done for MPH in Figure 2D)?

3. Exploratory findings for the effects of MPH on valence-specific reversal signals are reported in the OFC and DLPFC. It is not quite clear which area is considered OFC and DLPFC in Figure 4A. In any case, these clusters don't appear to be ventral enough to be located in the OFC. In fact, the clusters in the ventral PFC appear to be cut off. Is it possible that susceptibility artifacts and signal dropout prevented imaging in these areas? If this is just a masking issue, it may be worth re-analyzing these data with a mask that includes ventral prefrontal areas.

4. The analysis of the stimulus-specific activity in the visual cortex is an interesting way to test the gating hypothesis of striatal dopamine. I believe this hypothesis also predicts that functional connectivity between the striatum and sensory cortices should be selectively modulated. That is, connectivity with FFA should be enhanced for unexpected outcomes after face stimuli, whereas connectivity with PPA should be enhanced for unexpected outcomes after scene stimuli. Parallel effects have been reported for functional connections between the striatum and task-relevant subregions of PFC (Weber et al. 2018, PLOS Biology). It would be interesting to test whether dopaminergic drugs boost the functional connectivity between the striatum and sensory-specific cortices and whether these effects are modulated by dopamine synthesis capacity.

5. The discussion is quite long and may benefit from streamlining. In particular, the discussion on the drugs' mechanisms of action (lines 389-430) is (necessarily) quite speculative and could be rewritten more concisely.

6. SUL also acts as an antagonist at 5-HT_{1A} receptors. It would be important to discuss potential actions on these receptors, especially where results between MPH and SUL diverge. Could action on 5-HT_{1A} receptors explain the differential effects of MPH and SUL on overall accuracy (Figure 1D)?

Minor:

1. Are the effects reported in Figure 1B&C collapsed across all 3 sessions? This should be stated very clearly.

2. Please plot t- or z-scores instead of arbitrary beta values in Figure 1B and all other whole-brain statistical maps. Also, please use different colors to indicate $p < 0.05$ corrected and 0.001 uncorrected. The black outlines are difficult to see.

3. Line 994: Please report the uncorrected cluster-defining threshold that was used for whole-brain cluster-level FWE correction.

Reviewer #2 (Remarks to the Author):

This manuscript by van den Bosch et al. describes a well-designed, well-powered, and pre-registered pharmacological neuroimaging study that investigates the influence of two dopaminergic drugs (methylphenidate and sulpiride) on reversal learning and BOLD activation to unexpected outcomes in participants who vary as a function of striatal dopamine synthesis capacity (measured with FDOPA PET; Ki). Among other results, both drugs increased BOLD activation of the caudate, as well as the “stimulus-specificity” of BOLD response in visual cortical regions, following unexpected (“surprise”) outcomes, particularly for individuals with lower Ki in the caudate. Conversely, methylphenidate increased BOLD response in prefrontal cortical regions to unexpected reward vs. penalty information in individuals with higher Ki in the putamen. Finally, the two drugs had differential effects on the accuracy of trials following unexpected reward and unexpected penalty outcomes as a function of caudate Ki. At a high level, results of this work are of considerable interest to the field, particularly given the relatively large sample size for PET.

At the same time, there are nevertheless some important weaknesses in the paper. In particular, the paper is written in a way that is quite challenging to follow, which significantly undercuts the potential impact of the paper. Relatedly, there are questions about methods and results that need to be clarified.

Organization

- The organization of the paper is not adequate for the tremendous amount of data and large number of results. Having gone over the paper carefully several times, I think it is unrealistic to expect the average reviewer to be able to adequately appreciate and evaluate the paper in its current form. In particular, there is an over-reliance on supplemental materials to support and clarify claims in the primary text. This also leads to a lack of serious consideration of alternative interpretations and/or hypotheses, because it is not at all clear what one should or should not observe.

Methods

- Trial-level analyses: “First, we extracted the first eigenvariate ...from each cluster with significant dopamine-dependent drug effects in the univariate fMRI analysis.” Does this refer to all clusters in Table S3 that involve interaction with Ki (including caudate, putamen, and NAcc Ki)? Or to the subset that met corrected criteria? Was data each cluster added to a *separate* trial-wise model?

- Given known effects of age on dopamine system structure/function across the age range represented in this sample (e.g., Seaman et al., 2019, Hum Brain Mapp), the authors may wish to include age as a covariate in relevant behavioral and neuroimaging analyses (or to at least determine that it does not significantly influence effects of interest).

- I was a bit confused by the terms “dopamine-dependent” and “dopamine-independent” to refer to baseline-dependence/-independence of drug effects (as clearly drug effects are posited to be dopamine-related.) For clarity, the authors may consider changing these to something else, e.g. “baseline-dependent”/“baseline-independent”, throughout the manuscript.

- Were trial components (e.g., stimulus onset) other than outcome modeled in the primary GLM?

Results

- In certain parts of the text (e.g., section on PFC activation), analyses with putamen and nucleus accumbens ROI Ki values were explicitly described. In others, only analyses involving caudate Ki values are described. It’s not clear to me whether these additional (non-caudate) analyses were selectively conducted or just selectively reported. While I certainly understand the desire not to overburden an already complex manuscript with more data, if the authors are trying to make spatially-specific claims with respect to striatal dopamine, it would be important to present the full set of results (or to note whether lack of mention should be interpreted as a null result).

- It wasn’t clear to me why voxel-wise PET analyses were used in the section entitled “Striatal dopamine predicts drug effects on reversal learning.” This is perplexing as the authors immediately return to reporting ROI analyses and I could find no mention of spatial specificity gained from the voxel-wise analyses (the stated goal of voxel-wise analyses mentioned in the methods).

- In the section “Trial-level BOLD signal predicts better performance,” exactly which clusters were associated with accuracy? I don’t see an OFC cluster in Table S3 nor clearly pictured in any of the figures— does this refer to the bilateral anterior frontal region depicted in Fig S6B and/or one of the “aPFC” clusters listed in the table? (It is difficult to evaluate the claim in the discussion about valence-specific lateral OFC signal without this information.) Likewise, which striatal cluster (or ROI) was associated with accuracy? If it is the caudate region identified in “Methylphenidate enhances striatal BOLD signal” it would be useful to provide more information about this cluster (e.g., size).

- It would be useful to provide task RTs in addition to accuracy (e.g., in Table S1).

- In the Fig 2 legend, D corresponds to the clusters delineated in A (i.e., putamen) but the text description implies that it corresponds to the clusters shown in B (i.e., caudate).

- In “Striatal dopamine boosts stimulus-specificity in visual cortex,” the authors indicate that increased stimulus-specific activity was present for both scene and face comparisons but no estimates or statistics are provided. Was the magnitude of drug effect equivalent between the two visual categories/brain loci?

- Table S3: How are the $p=0$ values in various columns meant to be interpreted?

Reviewer #3 (Remarks to the Author):

This paper aims to answer important questions about how dopamine modulates reward learning, and interindividual differences underlying this. The main advance are evidence for the striatal gating hypothesis and for striatal dopamine synthesis capacity influencing the brain mechanisms underlying reversal learning. The study is well designed using a double blind, randomised, placebo controlled (with a double-dummy) design and a large sample size for a multimodal study. The imaging methods (multi echo fMRI and F-DOPA PET), image modelling and statistical analyses are methodologically sound. The main hypotheses and analysis methods were prespecified (albeit after 36 datasets were collected). The findings are interesting and the discussion is appropriate to the results. The main limitations are the potential for practice effects in the task and limitations of the BOLD signal as a measure of neuronal activity. The former can be tested for in additional analyses. I have some additional comments for clarification.

1. Abstract – suggest including the sample size in the abstract and data on the main effects

2. Introduction – Lines 111-113 – This statement should be referenced

3. Line 121 – the study being “well powered” is mentioned (here and in the discussion) but a power calculation is not reported in the main text, supplements or the preregistration. A large sample size is not enough for a study to be well powered as this also depends on the expected magnitude of the effect. Please provide a power calculation to support this statement

4. Methods – Changes in accuracy, reversal learning and BOLD signal may have been associated with practice effects as participants conducted the same task 3 times. Whilst order was randomised, it is not clear if this achieved balance across sessions. Please conduct additional analyses to test for order effects.

5. What was the method for randomisation and allocation concealment and specify if this included investigators as well in sessions and during image analysis?
6. Some psychiatric diagnoses could influence findings- in particular ADHD which is underdiagnosed in adults. It is not clear how this was assessed. Please explain and discuss the reliability of this.
7. The fMRI timings were designed to coincide with peak drug levels based on population data from other studies. There is interindividual variance in this. Were plasma levels of MPH and sulpiride obtained and how do they correspond to the fMRI timing? If they weren't obtained please add a discussion of this source of variance.
8. A functional connectivity parcellation of striatum was used for the PET but this is not explained and it seems anatomical ROIs are used in the analyses (caudate, putamen etc). Please provide more detail on the method and how this relates to the ROIs used in the analyses.
9. line 870 – how was the threshold of fewer than 20 reversals to exclude participants decided upon (this threshold is not pre-registered in the protocol). This could affect the generalisability of findings. Are the results sensitive to including these participants or to a different threshold?
10. Results. Please add the plots of the relationships in the main results rather than supplementary as these are the key data.
11. Line 160 - Effects of sulpiride on outcome related activity (the analysis which is reported for methylphenidate in figure 2A) are not reported, in the main text or in the supplements. Please add.
12. Figures – General comment on figures – the analysis included drug VS placebo effect on task BOLD signal, but the figures (eg 2A - “methylphenidate effect on BOLD signal during unexpected outcomes”) appear at first glance only to display the drug effect, not accounting for the placebo comparison. It would be clearer to include in the figure legends that this is versus placebo, eg “methylphenidate effect (MPH – PBO) on etc”
13. The distribution curves (eg in figures 2D, 2E) overlap and create a new colour, which is confusing. Can these be separated out or displayed differently?
14. Discussion. Line 307 – Please add a reference for methylphenidate being the most commonly used dopaminergic drug and further qualification for the context in which methylphenidate is the most commonly used dopaminergic drug (amphetamines are widely abused for example!)
15. “the non-specific performance 296 impairing effects of sulpiride were accompanied by task-nonspecific decreases in putamen signal 297 and increases in motor cortex signal (Figure S8D), possibly in line with an overall presynaptic 298 disinhibition of dopamine release, leading to decreased striatal indirect pathway activity and 299 disinhibition of motor responding, and resulting in a numerical, albeit not significant, overall 300 decrease in response times”. Here in the discussion (and also the intro), sulpiride's effects are discussed in terms of presynaptic effects but interpretation should also consider the post-synaptic effects, where substantial D2 occupancy is also expected at the dose used. Please add this discussion.
16. Figure 6– dopaminergic drug effects – clearer labelling needed to indicate that the top row refers to methylphenidate and sulpiride, whereas the bottom row refers only to methylphenidate.

17. Limitations of the BOLD signal in the context of the current study are relevant to the discussion - the drug-induced changes in the BOLD signal could be confounded by nonneuronal factors - eg drug induced changes in cerebral blood flow or cerebrovascular reactivity, which could produce (or mask) a BOLD response in the absence of a change in neural activity (Hum Brain Mapp. 2021 Jun 15;42(9):2766-2777). There is recent evidence for effects of DAT blockers on CBF linked to dopamine release (Sci Adv. 2021 Jun 9;7(24):eabg1512. doi: 10.1126/sciadv.abg1512.). Please add this consideration to the discussion.
18. The discussion should acknowledge that these were acute effects only, and that effects may vary with repeated drug administration (which is how these drugs are generally used in treatment of ADHD etc)
19. The discussion is rather too strong on concluding support for the gating hypothesis. It should acknowledge that causality has not been shown in this study
20. It is difficult to gauge the effect sizes for the effects. Please add a discussion of this, perhaps in terms of the amount of variance explained.

We would like to thank the reviewers for their helpful comments and critiques. We are grateful for the opportunity to clarify and improve the text, and for the suggestions for supplemental analyses.

Briefly, we have restructured and condensed parts of the discussion, added discussion points raised by the reviewers, and clarified figures and effect naming. We have also reported additional analyses investigating functional connectivity in the fMRI results and behavioral analyses controlling for effects of no interest. We believe that these changes have addressed all the concerns that were raised by the reviewers and improved the quality of our manuscript.

Our responses to the reviewers' points are in blue, with excerpts from the manuscript in italic. In the revised manuscript and supplemental information, we have highlighted the changes in yellow.

REVIEWER COMMENTS

Reviewer #1 (Remarks to the Author):

This human study tests the overall hypothesis that the effects of methylphenidate (MPH) on reversal learning performance depend on baseline levels of striatal dopamine and that these effects are driven by a gating mechanism, modulating neural responses to unexpected outcomes in stimulus-specific sensory cortex. The study combines dopamine pharmacology (catecholamine reuptake inhibitor MPH and D2/3 antagonist sulpiride [SUL]), task-based functional magnetic resonance imaging (fMRI), and positron emission tomography (PET) to measure dopamine (DA) synthesis capacity. The key result is that while MPH increased and SUL decreased behavioral performance, neural responses in the striatum to rewarding and punishing unexpected outcomes are enhanced by both MPH and SUL, particularly in subjects with low DA synthesis capacity. Paralleling these findings, both drugs enhanced responses to unexpected outcomes in stimulus-specific sensory cortex in a DA synthesis capacity-dependent way, providing support for the gating hypothesis of striatal DA. In addition, valence-specific responses to unexpected outcomes in the prefrontal cortex (PFC) were found to be modulated by MPH, particularly in subjects with high DA synthesis capacity in the striatum.

This is an extremely important study, addressing a long-standing and significant question in cognitive neuroscience. There are many strengths to this manuscript. The design is very strong with a large sample size (N=100), measurements of baseline DA synthesis capacity, and two different dopaminergic drugs, administered in a double-blind and placebo-controlled fashion. Event-related fMRI during a well-established reversal learning task is another strength. The manuscript is also well-written. The results will have a substantial impact on the field. I have a number of relatively minor comments suggestions, mostly to improve clarity.

We thank the reviewer for their appreciation of our work. We hope that the reviewer agrees with our modifications to improve the clarity of the manuscript.

Specific comments:

1. The results section does not always specify what comparisons are used to determine the effects of MPH and SUL. I assume that these are relative to placebo, but the existence of a placebo condition is not even mentioned before line 212. Are the behavioral (lines 132-136, Figure 1D-E) and neural effects (lines 161-165, Figure 2A-E, but also Figure 3A&B, Figure 4 B, and Figure 5B&D) of MPH and

SUL computed relative to placebo? Similarly, it should be made clear what is meant by “the drugs' effect” in line 181? This needs to be clarified throughout the manuscript.

The assumption of the reviewer is correct and we have added the following sentence at the beginning of the results section to clarify this (line 130): “... (all reported drug effects reflect a comparison of drug minus placebo).” In addition, we have explicitly clarified this in each figure’s caption by adding the statement “(relative to placebo)” after the first mention of drug effects. In line with a suggestion from Reviewer 3, we have also changed the title of most figure panels from e.g. “Methylphenidate effect on BOLD signal during unexpected outcomes” to “MPH – PBO: BOLD signal during unexpected outcomes”.

2. Why are the effects for SUL (Figure 2E) extracted from the cluster identified in Figure 2C, and not the cluster defined in Figure 2A (as it is done for MPH in Figure 2D)?

The effect of MPH extracted from the clusters in Figure 2A illustrates both the MPH-induced increase in signal across all participants AND the fact that this increase was stronger for participants with lower k_i^{cer} . For SUL there was no significant effect across all participants but only as a function of k_i^{cer} in the caudate nucleus cluster displayed in Figure 2C, hence we illustrated that effect in Figure 2E by extracting the values from that caudate nucleus cluster.

However, we agree that this discrepancy in the source of the values displayed in Figures 2D and 2E may be confusing. Therefore, we have changed Figure 2D to illustrate the k_i^{cer} -dependent effect of Figure 2B, so that the procedure for extraction is matched between the sulpiride vs placebo and methylphenidate vs placebo comparisons: Both Figures 2D and 2E display data extracted from the caudate nucleus cluster that surfaces as significant when assessing interactions between drug and k_i^{cer} at the voxel-level.

3. Exploratory findings for the effects of MPH on valence-specific reversal signals are reported in the OFC and DLPFC. It is not quite clear which area is considered OFC and DLPFC in Figure 4A. In any case, these clusters don’t appear to be ventral enough to be located in the OFC. In fact, the clusters in the ventral PFC appear to be cut off. Is it possible that susceptibility artifacts and signal dropout prevented imaging in these areas? If this is just a masking issue, it may be worth re-analyzing these data with a mask that includes ventral prefrontal areas.

The reviewer is correct that the signal from the very most rostro-ventral part of the PFC was truncated. This is indeed due to lack of signal in that area for several participants, which resulted in a reduced analysis mask at the group level.

The significant cluster containing the peak activity in the left hemisphere falls in Brodmann Area 10, the anterior prefrontal cortex (aPFC), as defined by the parcellation study of Sallet et al. (2013) and the wfupickatlas in SPM. The significant cluster in the right hemisphere lies close to BA 10, but the peak lies in the dIPFC (BA 46), as defined by Sallet et al. (2013). The dual-coded nature of the figure shows that while the significant clusters around the peaks (contours) lie in separate areas, the colored blobs (beta-values) show that the effect is located in both the aPFC and dIPFC in both hemispheres.

We have also corrected a mistake in the region labels in Table S3: the latter two regions labeled as aPFC / BA 10 labels should have been marked as dIPFC / BA 46.

While BA 10 is a relatively large area that includes the OFC, we agree that the significant cluster may not lie ventrally enough to justify the name OFC. Therefore, we have adjusted the naming in the result section to anterior prefrontal cortex “aPFC”.

We still think, however, that the OFC is involved in this contrast, not only due to the vicinity of the clusters to the OFC, but also due to the functional connectivity of the significant area with the OFC. For example, TMS stimulation of a brain area in the right hemisphere that overlaps with the activation shown in Figure 4A was shown to influence the OFC’s activity and associated behavioral changes (Howard et al., 2020). Therefore, we have retained the discussion of the involvement in the OFC but clarified this link in the discussion rather than naming the region OFC throughout the results section.

Howard, J. D., Reynolds, R., Smith, D. E., Voss, J. L., Schoenbaum, G., & Kahnt, T. (2020). Targeted Stimulation of Human Orbitofrontal Networks Disrupts Outcome-Guided Behavior. *Current Biology*, 30(3), 490-498.e4. <https://doi.org/10.1016/j.cub.2019.12.007>

Sallet, J., Mars, R. B., Noonan, M. P., Neubert, F.-X., Jbabdi, S., O’Reilly, J. X., Filippini, N., Thomas, A. G., & Rushworth, M. F. (2013). The Organization of Dorsal Frontal Cortex in Humans and Macaques. *Journal of Neuroscience*, 33(30), 12255–12274. <https://doi.org/10.1523/JNEUROSCI.5108-12.2013>

4. The analysis of the stimulus-specific activity in the visual cortex is an interesting way to test the gating hypothesis of striatal dopamine. I believe this hypothesis also predicts that functional connectivity between the striatum and sensory cortices should be selectively modulated. That is, connectivity with FFA should be enhanced for unexpected outcomes after face stimuli, whereas connectivity with PPA should be enhanced for unexpected outcomes after scene stimuli. Parallel effects have been reported for functional connections between the striatum and task-relevant subregions of PFC (Weber et al. 2018, PLOS Biology). It would be interesting to test whether dopaminergic drugs boost the functional connectivity between the striatum and sensory-specific cortices and whether these effects are modulated by dopamine synthesis capacity.

This is indeed an interesting supplemental analysis. We also thank the reviewer for the reference. We have now added this citation in the introduction (line 90). We have performed a psychophysiological interaction (PPI) analysis to test for task condition-dependent temporal correlation, a measure interpretable as functional connectivity, between BOLD signal in seed regions in the right caudate nucleus (defined as the clusters showing significant dopamine synthesis-dependent drug vs placebo effects) and in the individually defined FFA/PPA regions of interest. We tested whether there was significant change in the stimulus-specificity index of FFA/PPA signal as a function of right caudate nucleus activity during unexpected outcomes under methylphenidate or sulpiride compared with placebo. The analysis method and results are described in the Supplemental Information (lines 69-92 and 459-481) and is referred to in the Results section of the main manuscript (line 222-228).

This analysis revealed a significant dopamine synthesis-dependent effect of drug-induced increases in functional connectivity between the caudate nucleus and stimulus-specific visual association cortex after surprising outcomes for sulpiride, but not for methylphenidate. Thus, methylphenidate and sulpiride induced parallel increases in caudate nucleus signal and stimulus-specificity of signal in FFA/PPA in response to surprising outcomes for low compared with high dopamine synthesis participants (Figures 2&3), and the link between these effects was substantiated

for sulpiride by the PPI analysis uncovering functional connectivity between the caudate nucleus and FFA/PPA during surprising outcomes (Figure S6). This link was previously substantiated also for methylphenidate as a significant between-participants correlation between the effect of methylphenidate on caudate nucleus and FFA/PPA signal, rather than in terms of intra-individual links (lines 217-219). The difference between these links for each drug's effects is a recognized opponency in the statistical reliability of between vs within-subject effects (Hedge et al., 2018). A significant between-participants correlation for the effect of methylphenidate requires sufficient individual variability in the contrast estimates in the caudate nucleus and FFA/PPA. On the other hand, the reason that there is a significant effect of sulpiride within participants – i.e. low between-participants variability – is also the reason for the lack of a between-participants correlation.

Hedge, C., Powell, G., & Sumner, P. (2018). The reliability paradox: Why robust cognitive tasks do not produce reliable individual differences. *Behavior Research Methods*, 50(3), 1166–1186.
<https://doi.org/10.3758/s13428-017-0935-1>

5. The discussion is quite long and may benefit from streamlining. In particular, the discussion on the drugs' mechanisms of action (lines 389-430) is (necessarily) quite speculative and could be rewritten more concisely.

We agree with the reviewer that the speculations on the drugs' mechanisms of action was lengthy. Accordingly, we have edited that section on the mechanisms to condense it (lines 404-439), and we moved the preceding paragraph to after the discussion on the mechanism and edited it in an attempt to increase reading flow (new lines 449-460).

6. SUL also acts as an antagonist at 5-HT1A receptors. It would be important to discuss potential actions on these receptors, especially where results between MPH and SUL diverge. Could action on 5-HT1A receptors explain the differential effects of MPH and SUL on overall accuracy (Figure 1D)?

Sulpiride action on 5-HT1A receptors becomes substantial at high doses, due to the drug's low affinity to the receptor (Yonemura et al., 1998). It seems plausible that at the lower dose of 400 mg used here, there would have been some action on the 5-HT1A receptors as well. Following the general principle of biphasic modes of action of neuromodulator drugs at pre- and postsynaptic receptors, the low dose would likely have mostly presynaptic effects, resulting in increased serotonin release. We have now acknowledged this point in the discussion (lines 498-502): "*A second consideration is the fact that sulpiride also acts as an antagonist of 5-HT1A receptors. However, we consider it less plausible that the effects observed here reflect changes in serotonin receptor activity, because such effects would be more substantial only at higher doses of sulpiride, due to its low affinity for the 5-HT1A receptor [Yonemura et al., 1998].*"

Yonemura, K., Miyanaga, K., & Machiyama, Y. (1998). Profiles of the affinity of antipsychotic drugs for neurotransmitter receptors and their clinical implication. *The KITAKANTO Medical Journal*, 48(2), 87–102.

Minor:

1. Are the effects reported in Figure 1B&C collapsed across all 3 sessions? This should be stated very clearly.

The effects in Figure 1B&C are indeed collapsed across all 3 sessions. To make that clearer, we have added the text “*Across all three sessions, ...*” to the beginning of line 125, and to the figure legends of Figures 1B&C (line 149, and 154).

2. Please plot t- or z-scores instead of arbitrary beta values in Figure 1B and all other whole-brain statistical maps. Also, please use different colors to indicate $p < 0.05$ corrected and 0.001 uncorrected. The black outlines are difficult to see.

We agree that the black outlines indicating significant clusters were suboptimal for some of the figures. We have changed it to black outlines for red blobs and white outlines for blue blobs, because the black outlines were least visible around blue blobs.

For visualization, we have used a dual-coded approach displaying both the t-values and beta values. The blue-black and red-yellow colors code the beta values. The t-values are displayed in terms of the opaqueness of the colored blobs. All blobs that pass the threshold for significance are fully opaque; the rest are more transparent the lower the associated t-value and fully transparent below a t-value of 1 (to reduce clutter). We have not changed this dual-coded nature of the whole-brain maps, because we feel that visualizing the data in this way is more revealing than the more commonly used approach of only displaying t-values, which hides all effects that do not reach the cut-off score for statistical significance (see also Allen et al., 2012).

This explanation of the voxel-wise result figures was provided in the legend of Figure 1B, but we have now also added this to the methods section (lines 1043-1049).

Allen, E. A., Erhardt, E. B., & Calhoun, V. D. (2012). Data Visualization in the Neurosciences: Overcoming the Curse of Dimensionality. *Neuron*, 74(4), 603–608.
<https://doi.org/10.1016/j.neuron.2012.05.001>

3. Line 994: Please report the uncorrected cluster-defining threshold that was used for whole-brain cluster-level FWE correction.

We have added this information in the new clause in line 1039: “*..., with an initial uncorrected cluster-defining threshold of $p < 0.001$.*”

Reviewer #2 (Remarks to the Author):

This manuscript by van den Bosch et al. describes a well-designed, well-powered, and pre-registered pharmacological neuroimaging study that investigates the influence of two dopaminergic drugs (methylphenidate and sulpiride) on reversal learning and BOLD activation to unexpected outcomes in participants who vary as a function of striatal dopamine synthesis capacity (measured with FDOPA PET; Ki). Among other results, both drugs increased BOLD activation of the caudate, as well as the “stimulus-specificity” of BOLD response in visual cortical regions, following unexpected (“surprise”) outcomes, particularly for individuals with lower Ki in the caudate. Conversely, methylphenidate increased BOLD response in prefrontal cortical regions to unexpected reward vs. penalty information in individuals with higher Ki in the putamen. Finally, the two drugs had differential effects on the accuracy of trials following unexpected reward and unexpected penalty outcomes as a function of

caudate Ki. At a high level, results of this work are of considerable interest to the field, particularly given the relatively large sample size for PET.

At the same time, there are nevertheless some important weaknesses in the paper. In particular, the paper is written in a way that is quite challenging to follow, which significantly undercuts the potential impact of the paper. Relatedly, there are questions about methods and results that need to be clarified.

We thank the reviewer for their effort and considerations and are happy they found our results to be of interest. We hope the reviewer agrees that the revised manuscript is improved in clarity and that their concerns were adequately addressed.

Organization

- The organization of the paper is not adequate for the tremendous amount of data and large number of results. Having gone over the paper carefully several times, I think it is unrealistic to expect the average reviewer to be able to adequately appreciate and evaluate the paper in its current form. In particular, there is an over-reliance on supplemental materials to support and clarify claims in the primary text. This also leads to a lack of serious consideration of alternative interpretations and/or hypotheses, because it is not at all clear what one should or should not observe.

We agree that the paper covers a large dataset and many results. The Supplemental Information contains supporting figures, additional task performance data, and control analyses, together with additional statistics for significant interaction effects in the main text broken down into their constituent parts. We realize that the Supplemental Information is extensive, but the main claims and results are described in the main text. We tried to keep the supplement digestible by aligning the order of the sections to the order of the main text as best as possible, and with clear references in the main text to accompanying support in the supplement.

The strongest reliance of the main results on supplemental data was arguably the section on the response time results. To reduce the reliance on the Supplemental Information, we have moved the description and accompanying figure of the significant drug effects on response times from the supplement to the main text (lines 294-312; Figure 6), and adapted the references to the figure and text accordingly throughout the text.

One reason for the extent of the Supplemental Information is that it contains many scatter plot figures that are merely elaborated versions of the median-split versions of the figures in the main text, thus displaying the same result but in an alternative manner. We think this approach does justice to the need to both clearly communicate the results in the main text as well as give the readers access to the full data points (see also comment 10 of Reviewer 3). If the reviewer and/or editor prefer that we move the scatter plots to the main text, then we will do so.

To make the organization of the paper's main text and supplemental information clear to readers, we now refer to the overview figure of the main results (Figure 7) at the beginning of the Results section (line 138) and we provide a bullet point list of the contents of the Supplemental Information at the end of the Results section in the main text (lines 313-326):

"In the Supplemental Information we provide:

- *descriptive data of dopamine synthesis capacity and results from validating analyses of general task performance and fMRI effects*
- *additional figures and statistics pertaining to breakdowns of significant interaction effects described in the main text. Many figures presented in the main text are median-split to clearly show an effect; in the Supplemental Information, corresponding figures show those effects in a scatter plot version for complete reporting.*
- *the results of two supplemental control analyses: (i) to correct behavioral effects for age and session number, and (ii) to establish that observed drug effects are not explained by effects on win-stay/lose-shift behavior.*
- *the new results of an exploratory generalized psychophysiological interaction (gPPI) analysis investigating drug effects on functional connectivity between brain areas observed in the main results.”*

Methods

- Trial-level analyses: “First, we extracted the first eigenvariate ...from each cluster with significant dopamine-dependent drug effects in the univariate fMRI analysis.” Does this refer to all clusters in Table S3 that involve interaction with Ki (including caudate, putamen, and NAcc Ki)? Or to the subset that met corrected criteria? Was data each cluster added to a **separate** trial-wise model?

We thank the reviewer for pointing us to this section, which indeed required some changes to improve clarity and consistency. We ran trial-level analysis models with BOLD signal predictors from the aPFC cluster (Figure 4A), the putamen clusters where there was a significant effect of methylphenidate on surprising outcome signal (Figure 2A), and estimates of stimulus-specificity of BOLD signal in the individually defined FFA/PPA. Additionally, we included trial-level BOLD signal from the bilateral cluster in the supramarginal gyrus where there was a dopamine-independent main effect of methylphenidate (Supplemental Table S3) to use as control region. We have now also added analyses of trial-level BOLD estimates in the caudate nucleus clusters where there were dopamine synthesis-dependent effects of methylphenidate and sulpiride (Figure 2B-C). We have moved the results and methods for these analyses to the Supplemental Information (lines 222-255 and 482-515; see also the answer to the third ‘Results’ comment, below). We ran separate trial-wise models for each cluster.

- Given known effects of age on dopamine system structure/function across the age range represented in this sample (e.g., Seaman et al., 2019, Hum Brain Mapp), the authors may wish to include age as a covariate in relevant behavioral and neuroimaging analyses (or to at least determine that it does not significantly influence effects of interest).

This is a valid point. We have included age and session number (see comment Reviewer 3) as covariates in control analyses, which revealed that they did not influence the drug effects on accuracy. Only the effect of methylphenidate on reward versus punishment prediction speed was no longer significant after including the extra variables, however there were also no significant effects of age and session number. This information has been added to the Supplemental Information in lines 174-180 and lines 208-215.

- I was a bit confused by the terms “dopamine-dependent” and “dopamine-independent” to refer to baseline-dependence/-independence of drug effects (as clearly drug effects are posited to be

dopamine-related.) For clarity, the authors may consider changing these to something else, e.g. “baseline-dependent”/“baseline-independent”, throughout the manuscript.

We understand the confusion in using the terms “dopamine-(in)dependent”. We have changed the phrase to “dopamine synthesis-dependent” throughout the manuscript.

- Were trial components (e.g., stimulus onset) other than outcome modeled in the primary GLM?

In the primary GLM, the outcome onsets of the four different outcome types (separated by their association with the face/scene stimulus) were the only trial components in the model.

Stimulus onset was the only other trial component used in another GLM, namely in the model that was used to localize the FFA and PPA per individual participant.

Results

- In certain parts of the text (e.g., section on PFC activation), analyses with putamen and nucleus accumbens ROI K_i values were explicitly described. In others, only analyses involving caudate K_i values are described. It's not clear to me whether these additional (non-caudate) analyses were selectively conducted or just selectively reported. While I certainly understand the desire not to overburden an already complex manuscript with more data, if the authors are trying to make spatially-specific claims with respect to striatal dopamine, it would be important to present the full set of results (or to note whether lack of mention should be interpreted as a null result).

We agree that it should have been made more explicit that the lack of mention of effects of putamen and nucleus accumbens dopamine synthesis should be interpreted as a null result. In the introduction we note that we focused our analyses on the cognitive subregion of the striatum, the caudate nucleus, and additionally explored the role of dopamine synthesis capacity in the putamen and nucleus accumbens. Hence, the results described in the section on PFC activation, which depended not on caudate nucleus but on putamen dopamine, are exploratory. We have now stated this approach also explicitly in the methods section (line 1000): “*Our confirmatory analyses focused on dopamine synthesis capacity in the caudate nucleus, and we additionally explored the effect of dopamine synthesis capacity in the putamen and nucleus accumbens.*”. In addition, we now also mention the null results for the other striatal ROIs in each Results section (lines 176, 215, 271, 296).

- It wasn't clear to me why voxel-wise PET analyses were used in the section entitled “Striatal dopamine predicts drug effects on reversal learning.” This is perplexing as the authors immediately return to reporting ROI analyses and I could find no mention of spatial specificity gained from the voxel-wise analyses (the stated goal of voxel-wise analyses mentioned in the methods).

We agree that the current wording of that section suggests a reliance on the voxel-wise analysis for the inference. The purpose of the voxel-wise analysis was simply to visualize the effect of dopamine synthesis capacity. We have modified the sentence stating this goal in the methods (line 1002) from “*For precise localization and illustration of the effects ...*” to “*For visualization of the effects ...*”.

To make it clearer that the inference is based on the ROI analyses, we have rephrased the section “Striatal dopamine predicts drug effects on reversal learning” (line 264-275), ending with the sentence (line 274): “*The significant effects are visualized in Figures 5B and D, using voxel-wise PET*

analyses with the behavioral drug effect as individual difference predictor.” We also swapped Figures 5B,D with 5A,C, so the figures corresponding to the ROI analysis are presented first.

- In the section “Trial-level BOLD signal predicts better performance,” exactly which clusters were associated with accuracy? I don’t see an OFC cluster in Table S3 nor clearly pictured in any of the figures— does this refer to the bilateral anterior frontal region depicted in Fig S6B and/or one of the “aPFC” clusters listed in the table? (It is difficult to evaluate the claim in the discussion about valence-specific lateral OFC signal without this information.) Likewise, which striatal cluster (or ROI) was associated with accuracy? If it is the caudate region identified in “Methylphenidate enhances striatal BOLD signal” it would be useful to provide more information about this cluster (e.g., size).

We realize that the naming of the prefrontal clusters in the text versus Table S3 was different. We now refer to the prefrontal clusters as “aPFC” instead of OFC in the main text (see also comment reviewer 1), which aligns it with the naming in Table S3. The bilateral cluster depicted in Figure 4A (overlaps with the similar clusters in Figure S6B) was used in the trial-wise analysis and positively predicted accuracy.

In addition to the aPFC cluster, we used trial-level estimates of BOLD signal in the putamen clusters where there was a significant effect of methylphenidate on surprising outcome signal (Figure 2A), as well as estimates of stimulus-specificity of BOLD signal in the individually defined FFA/PPA. Outcome-related signal in the aPFC, putamen and FFA/PPA positively predicted subsequent accuracy and response speed. The control region in the supramarginal gyrus did not predict accuracy, only response times.

We have now also added analyses of trial-level BOLD estimates in the caudate nucleus clusters where there were dopamine synthesis-dependent effects of methylphenidate and sulpiride. However, in contrast to the putamen, signal in the newly analyzed caudate nucleus clusters predicted worse subsequent accuracy (methylphenidate only) and slower response times (both for methylphenidate and sulpiride). While the opposite effects on response times may reflect the differential involvement of the putamen and caudate nucleus in motor and cognitive processes, respectively, the result that caudate nucleus signal negatively predicted accuracy is unexpected. It may be that greater caudate nucleus activity hampers performance on regular trials and only benefits reversal trials. The direction of the interaction effect of caudate signal with unexpected versus expected outcomes is in line with this, but it was not significant.

While these new findings for the caudate nucleus clusters do not invalidate the previously reported finding for the putamen, combined they do make the results of these trial-wise analyses less clear. Given that these analyses are extra explorations, we have decided to move this section to the Supplemental Information. There, the description of the results is in lines 222-255, the updated methods section is in lines 482-515, and we have added a section to the supplemental discussion about the surprising caudate results in lines 316-342. In the updated methods section, we now also provide the size of each cluster in number of voxels (aPFC: 46 voxels; putamen: 101 voxels; MPH right caudate nucleus: 18 voxels; SUL right caudate nucleus: 15 voxels; supramarginal gyrus: 124 voxels).

- It would be useful to provide task RTs in addition to accuracy (e.g., in Table S1).

We thank the reviewer for this suggestion; we have added the mean RTs to Table S1.

- In the Fig 2 legend, D corresponds to the clusters delineated in A (i.e., putamen) but the text description implies that it corresponds to the clusters shown in B (i.e., caudate).

Figure 2D displayed the effect of MPH extracted from the clusters in Figure 2A. That way the figure illustrated both the MPH-induced increase in signal across all participants AND the fact that this increase was stronger for participants with lower k_i^{cer} . For SUL there was no effect across all participants but only as a function of k_i^{cer} in the caudate cluster displayed in Figure 2C, hence we illustrated that effect in Figure 2E by extracting the values from that caudate cluster.

However, we agree that this discrepancy in the source of the values displayed in Figures 2D and 2E may be confusing (see also comment 2 of Reviewer 1), and that the text is more focused on the k_i^{cer} -dependent effect as displayed in Figure 2B. Therefore, we have changed Figure 2D to illustrate the k_i^{cer} -dependent effect of Figure 2B, so that the procedure for extraction is matched between the sulpiride vs placebo and methylphenidate vs placebo comparisons: Both Figures 2D and 2E display data extracted from the caudate cluster that surfaces as significant when assessing interactions between drug and k_i^{cer} at the voxel-level.

- In “Striatal dopamine boosts stimulus-specificity in visual cortex,” the authors indicate that increased stimulus-specific activity was present for both scene and face comparisons but no estimates or statistics are provided. Was the magnitude of drug effect equivalent between the two visual categories/brain loci?

The statistics for the full interactions are provided for MPH and SUL in the legend of Figure 3.

These represent the drug effects on the stimulus-specificity index of relative FFA/PPA signal during the presentation of unexpected outcomes compared with expected outcomes. To create the index, we first subtracted signal related to outcomes associated with the scene stimulus from signal related to outcomes associated with the face stimulus (face – scene) in both the FFA and PPA. The index is created by subtracting the resulting value in the PPA from that in the FFA (FFA – PPA). For full reporting, we have now added the statistics for the drug effects separated by FFA/PPA ROI (face – scene signal in each ROI) in the Supplemental Information (line 56-63). The drug effects were of similar magnitude (but opposite sign) in the FFA and PPA. However, the interaction MPH x expectancy x caudate k_i^{cer} was not significant in the FFA or PPA in isolation, only the difference between these regions was significant. The interaction effect SUL x expectancy x caudate k_i^{cer} was significant in the FFA but not the PPA (FFA: $F_{(1,80)} = 4.21$, $p = 0.044$; PPA: $F_{(1,80)} = 1.77$, $p = 0.187$).

To make it clear that the text in the results section of the main manuscript pertains to the full interaction term, we have added the brackets explaining that the scene-related activity in PPA is relative to the face activity and the FFA in line 209: “Conversely, the drugs enhanced signal in the PPA (compared with FFA) when the unexpected outcome was associated with the scene stimulus (compared with the face).”

- Table S3: How are the p=0 values in various columns meant to be interpreted?

The p=0 values were automatically rounded to zero. We have now replaced those values with “<0.001”. In addition, we have removed the column “p(z)”. This column (was all zeroes) indicated uncorrected voxel p-values, but was uninformative as the initial uncorrected threshold for analysis was set at $p < 0.001$.

Reviewer #3 (Remarks to the Author):

This paper aims to answer important questions about how dopamine modulates reward learning, and interindividual differences underlying this. The main advance are evidence for the striatal gating hypothesis and for striatal dopamine synthesis capacity influencing the brain mechanisms underlying reversal learning. The study is well designed using a double blind, randomised, placebo controlled (with a double-dummy) design and a large sample size for a multimodal study. The imaging methods (multi echo fMRI and F-DOPA PET), image modelling and statistical analyses are methodologically sound. The main hypotheses and analysis methods were prespecified (albeit after 36 datasets were collected). The findings are interesting and the discussion is appropriate to the results. The main limitations are the potential for practice effects in the task and limitations of the BOLD signal as a measure of neuronal activity. The former can be tested for in additional analyses. I have some additional comments for clarification.

We thank the reviewer for their effort and helpful suggestions to improve the manuscript. We hope they agree with the changes made.

1. Abstract – suggest including the sample size in the abstract and data on the main effects

We have added the sample size information to the abstract in line 38: *“Young healthy adults (N=100) were scanned...”*.

We have also added a sentence on the main effects (in line 41: *“Methylphenidate improved and sulpiride decreased overall accuracy and response speed.”*), and slightly modified the following sentence to clarify the parallel effects of both drugs on reward/punishment learning (line 42: *“Both drugs boosted ...”*).

In order to comply with the limit of 150 words in the abstract, we have adapted the first and last sentences of the abstract: removed clause *“..., and there are concerns about the potential for abuse.”* in line 34, and changed line 45 to *“These results unravel the mechanisms by which methylphenidate gates both attention and reward learning.”*

2. Introduction – Lines 111-113 – This statement should be referenced

This comment pertains to *“... the pervasive hypothesis that methylphenidate’s effects depend on baseline levels of striatal dopamine”*. We have now referenced the statement with reference 45:

Cools, R. & D’Esposito, M. Inverted-U-Shaped Dopamine Actions on Human Working Memory and Cognitive Control. *Biological Psychiatry* 69, e113–e125 (2011).

3. Line 121 – the study being “well powered” is mentioned (here and in the discussion) but a power calculation is not reported in the main text, supplements or the preregistration. A large sample size is not enough for a study to be well powered as this also depends on the expected magnitude of the effect. Please provide a power calculation to support this statement

We agree with the reviewer that a large sample size does not necessarily equal high statistical power. There is no easy formula to determine statistical power for Bayesian mixed-effects models. It would be possible to run simulations with varying sample sizes to get at a similar estimate, but because we have not performed those a priori, we have changed the statements of “well-powered” to “large sample” in lines 120 and 386.

4. Methods – Changes in accuracy, reversal learning and BOLD signal may have been associated with practice effects as participants conducted the same task 3 times. Whilst order was randomised, it is not clear if this achieved balance across sessions. Please conduct additional analyses to test for order effects.

The drug order across sessions was close, but not perfectly balanced, across the 88 participants in the final analysis. To assess order effects, we have run additional mixed-effects models that included session number as an additional variable that interacted with outcome expectancy, outcome valence, and dopamine synthesis capacity. These control analyses additionally included age as a potentially confounding factor (see comment Reviewer 1). The control analyses revealed that age and session number did not influence the drug effects on accuracy. Only the effect of methylphenidate on reward versus punishment prediction speed was no longer significant after including the extra variables, although there were also no significant effects of age and session number. This information has been added to the Supplemental Information in lines 174-180 and lines 208-215.

5. What was the method for randomisation and allocation concealment and specify if this included investigators as well in sessions and during image analysis?

The randomization of drug administration order for 100 participants was performed prior to the onset of the study by an independent researcher not involved in the project. Individual participants were assigned a participant number in chronological order, separated by gender (1-50 female, 51-100 male). Preparation of the medication in accordance with the prespecified administration order per participant number was performed by the pharmacy and blinded before delivery to the researchers involved in data collection. Thus, both participants and researchers were blind to drug administration order during data collection.

During final image analyses, the researchers were not blind to drug status. Due to other parts of the overarching project moving faster in the analysis and publication process, unblinding of drug status was necessary before image analysis of the current task data was complete.

6. Some psychiatric diagnoses could influence findings- in particular ADHD which is underdiagnosed in adults. It is not clear how this was assessed. Please explain and discuss the reliability of this.

Before inclusion in the study proper, a participant's eligibility was assessed during an intake session. The assessment included a systematic psychiatric screening interview (M.I.N.I. Plus 5.0.0), which also contains a section pertaining to ADHD symptoms during both childhood and adulthood. Any indication for the presence of ADHD currently or in childhood was reason for exclusion, although this did not happen in our sample.

More details regarding inclusion criteria, as well as a figure displaying self-reported ADHD scores, are available in the preprint describing the design and methods of the overarching projects (was ref 93, now ref 98). We have now added a reference to this document in the participants section of the methods in the main text (line 809), and we have updated the reference itself in the manuscript to correctly point to the document on OSF (reference pasted below).

Määttä, J. I., van den Bosch, R., Papadopetraki, D., Hofmans, L., Lambregts, B., Westbrook, A., Verkes, R.-J., & Cools, R. (2021). Predicting effects of methylphenidate and sulpiride on brain and cognition: A pharmaco-fMRI, PET study. Design and descriptives. Preprint OSF. <https://doi.org/10.31219/osf.io/d3h8e>

7. The fMRI timings were designed to coincide with peak drug levels based on population data from other studies. There is interindividual variance in this. Were plasma levels of MPH and sulpiride obtained and how do they correspond to the fMRI timing? If they weren't obtained please add a discussion of this source of variance.

We did not obtain plasma levels of methylphenidate and sulpiride. We have now mentioned this source of variance in the discussion in line 495: *"In addition, the current design with fixed timings between drug intake and task performance, and without monitoring of drug plasma levels, did not enable us to control for interindividual differences in drug plasma concentrations during scanning."*

8. A functional connectivity parcellation of striatum was used for the PET but this is not explained and it seems anatomical ROIs are used in the analyses (caudate, putamen etc). Please provide more detail on the method and how this relates to the ROIs used in the analyses.

The ROI masks were obtained from an independent functional connectivity parcellation project that we carried out previously (Piray et al., 2017, Cereb Cortex). We have adapted the relevant sentences in the Methods section to the following (lines 966-971):
"The striatal ROI masks were obtained from an independent, functional connectivity-based parcellation of the striatum that we conducted in a previous study¹¹ (Figure S15). That study used a clustering method to identify five striatal subregions based on resting state functional connectivity, which we combined into our three ROI masks. These masks map well onto the anatomical structures of the caudate nucleus, putamen and nucleus accumbens."

Thus, we did not use anatomical ROIs, but the functional connectivity-based ROIs map well onto the anatomical structures.

Piray, P., Ouden, D., E.M, H., Schaaf, V. D., E, M., Toni, I., & Cools, R. (2017). Dopaminergic Modulation of the Functional Ventrodorsal Architecture of the Human Striatum. *Cerebral Cortex*, 27(1), 485–495. <https://doi.org/10.1093/cercor/bhv243>

9. line 870 – how was the threshold of fewer than 20 reversals to exclude participants decided upon (this threshold is not pre-registered in the protocol). This could affect the generalisability of findings. Are the results sensitive to including these participants or to a different threshold?

We agree with the reviewer that the threshold would preferably have been included in the preregistration. While we considered and determined the threshold after uploading the registration, we did so before completion of the data collection and before starting data analyses.

We considered fewer than 20 reversal trials a sign of task disengagement. To test this, we correlated the number of reversals in a session with the number of missed trials in that session, across all available sessions (i.e. without any exclusions). This revealed a strong negative correlation ($t_{(291)} = -17.49$, $p < 2.2e-16$), indicating that those sessions on which participants had few reversal trials also contained a large number of missed trials (rather than simply incorrect trials), presumably reflecting task disengagement. The figure below illustrates this correlation and shows that the preset cutoff of 20 reversals resulted in exclusion of mostly sessions with many missed trials.

We have not run additional models that included these participants to check for sensitivity of the results to this threshold. Such work would further increase the number of analyses, but if the editor and/or reviewer feel strongly about this point we can add these additional models.

Horizontal line indicates mean + 1 SD of number of misses.

10. Results. Please add the plots of the relationships in the main results rather than supplementary as these are the key data.

We are not sure to which plots the reviewer is referring here. The key effects/relationships are presented in the main text as median-split plots, and again in the supplementary text, as scatter plots. We think this approach does justice to the need to both clearly communicate the results as well as report the data completely. If the reviewer and/or editor prefer that we move the scatter plots to the main text, then we will do so.

11. Line 160 - Effects of sulpiride on outcome related activity (the analysis which is reported for methylphenidate in figure 2A) are not reported, in the main text or in the supplements. Please add.

We have added a sentence to the final paragraph of “Methylphenidate enhances striatal BOLD signal” to explain that there was no effect of sulpiride across participants (line 177): “... there were no effects of sulpiride on BOLD signal to unexpected outcomes when dopamine synthesis capacity was not taken into account.”

12. Figures – General comment on figures – the analysis included drug VS placebo effect on task BOLD signal, but the figures (eg 2A - “methylphenidate effect on BOLD signal during unexpected outcomes”) appear at first glance only to display the drug effect, not accounting for the placebo comparison. It would be clearer to include in the figure legends that this is versus placebo, eg “methylphenidate effect (MPH – PBO) on etc”

In response to a comment of Reviewer 1, we have added the statement “(relative to placebo)” at the first mention of drug effects in each figure caption. We have now also changed the title of most figure panels from e.g. “Methylphenidate effect on BOLD signal during unexpected outcomes” to “MPH – PBO: BOLD signal during unexpected outcomes”.

13. The distribution curves (eg in figures 2D, 2E) overlap and create a new colour, which is confusing. Can these be separated out or displayed differently?

It is unfortunate that the Reviewer found the figures confusing. We think that the different shade of color where the distributions overlap is in fact informative. Particularly in figures such as 2D and 2E, where there is large overlap. It illustrates the lack of difference between the two conditions. This would become less apparent if the distributions would be separated.

14. Discussion. Line 307 – Please add a reference for methylphenidate being the most commonly used dopaminergic drug and further qualification for the context in which methylphenidate is the most commonly used dopaminergic drug (amphetamines are widely abused for example!)

We have changed the final clause in the first sentence of the discussion (line 333 to: “..., which is the most prescribed dopaminergic drug together with the related amphetamines (<https://clincalc.com/DrugStats/>).”

15. “the non-specific performance 296 impairing effects of sulpiride were accompanied by task-nonspecific decreases in putamen signal 297 and increases in motor cortex signal (Figure S8D), possibly in line with an overall presynaptic 298 disinhibition of dopamine release, leading to decreased striatal indirect pathway activity and 299 disinhibition of motor responding, and resulting in a numerical, albeit not significant, overall 300 decrease in response times”. Here in the discussion (and also the intro), sulpiride’s effects are discussed in terms of presynaptic effects but interpretation should also consider the post-synaptic effects, where substantial D2 occupancy is also expected at the dose used. Please add this discussion.

We agree with the Reviewer that addressing potential postsynaptic effects of sulpiride was missing in the discussion. We have now added the following to the discussion in the main text in lines 416-418:

“While the relatively low dose of sulpiride used here is also consistent with presynaptic action³⁴, [Naef et al., 2017], we cannot exclude the possibility that it acted also postsynaptically, perhaps in line with the overall reduction in accuracy across participants [Eisenegger et al., 2014].”

Naef, M., Müller, U., Linssen, A., Clark, L., Robbins, T. W., & Eisenegger, C. (2017). Effects of dopamine D2/D3 receptor antagonism on human planning and spatial working memory. *Translational Psychiatry*, 7(4), e1107. <https://doi.org/10.1038/tp.2017.56>

Eisenegger, C., Naef, M., Linssen, A., Clark, L., Gandamaneni, P. K., Müller, U., & Robbins, T. W. (2014). Role of Dopamine D2 Receptors in Human Reinforcement Learning. *Neuropsychopharmacology*, 39(10), 2366–2375. <https://doi.org/10.1038/npp.2014.84>

16. Figure 6– dopaminergic drug effects – clearer labelling needed to indicate that the top row refers to methylphenidate and sulpiride, whereas the bottom row refers only to methylphenidate.

We thank the reviewer for highlighting that it should be clearer which panel illustrates which drugs. The bottom left panel displays the dopamine synthesis-dependent drug effect on behavior that is present for both methylphenidate and sulpiride; the bottom right panel shows the drug effect on prefrontal BOLD signal that is only present for methylphenidate.

To make this clearer, we have added an inset to each panel with the text “Drugs: MPH & SUL” or “Drugs: MPH”, and we have adapted the figure caption (now Figure 7; lines 441-448):

“The dopaminergic drugs methylphenidate and sulpiride boosted outcome surprise signals in the caudate nucleus and stimulus-specific visual association cortex in ‘low striatal dopamine’ participants (top row panels). This was accompanied by the drugs boosting punishment compared with reward-based reversal learning in the ‘low striatal dopamine’ participants compared with the ‘high striatal dopamine’ participants (bottom left panel). By contrast, the drugs boosted relative reward-based reversal learning to a greater degree in ‘high striatal dopamine’ participants (bottom left panel), and methylphenidate also boosted the associated prefrontal BOLD signal to a greater degree in ‘high striatal dopamine’ participants (bottom right panel). MPH: methylphenidate; SUL: sulpiride.”

17. Limitations of the BOLD signal in the context of the current study are relevant to the discussion - the drug-induced changes in the BOLD signal could be confounded by nonneuronal factors - eg drug induced changes in cerebral blood flow or cerebrovascular reactivity, which could produce (or mask) a BOLD response in the absence of a change in neural activity (Hum Brain Mapp. 2021 Jun 15;42(9):2766-2777). There is recent evidence for effects of DAT blockers on CBF linked to dopamine release (Sci Adv. 2021 Jun 9;7(24):eabg1512. doi: 10.1126/sciadv.abg1512.). Please add this consideration to the discussion.

The Reviewer is right to raise this point of discussion. We have added this consideration to the discussion in the manuscript as follows (lines 502-510):

“Finally, another important issue to consider in all pharmacological fMRI studies is the degree to which any of the BOLD changes of interest might reflect non-neuronal drug effects, such as changes in cerebral blood flow or cerebrovascular reactivity in the absence of changes in neural activity [Hawkins et al., 2021. Hum Brain Mapp. 2021 Jun 15;42(9):2766-2777]. We consider this unlikely in this case, because the effects were regionally selective and task specific. Specifically, the effect of methylphenidate on striatal BOLD signal was observed during unexpected relative to expected outcomes. Moreover, it was observed only in the striatum, and did not extend to other regions activated by the unexpected outcomes (Figure 1B). Similarly, the effect of methylphenidate on aPFC BOLD signal was observed during unexpected reward relative to punishment outcomes.”

18. The discussion should acknowledge that these were acute effects only, and that effects may vary with repeated drug administration (which is how these drugs are generally used in treatment of ADHD etc)

We agree, and have added this acknowledgement in the discussion (line 490-495):

“It is important to note that the current results reflect effects of acute administrations of fixed doses of methylphenidate and sulpiride, whereas these drugs are typically taken repeatedly as long-term treatment of ADHD and psychosis in a variety of doses. Thus, the current results do not address potential long-term effects on brain and behavior, and higher doses might produce different effects, particularly for sulpiride, which elicits stronger postsynaptic effects at higher doses [Naef et al., 2017].”

Naef, M., Müller, U., Linssen, A., Clark, L., Robbins, T. W., & Eisenegger, C. (2017). Effects of dopamine D2/D3 receptor antagonism on human planning and spatial working memory. *Translational Psychiatry*, 7(4), e1107. <https://doi.org/10.1038/tp.2017.56>

19. The discussion is rather too strong on concluding support for the gating hypothesis. It should acknowledge that causality has not been shown in this study

To avoid the suggestion of claiming causal evidence for the hypothesis, we have made two adaptations to the relevant paragraph in the discussion: (1) in line 373 we replaced “*established the output gating hypothesis*” with “*provide support for the output gating hypothesis*”, and (2) in line 396 we have added a clause to the start of the final sentence “*While the current experiment cannot demonstrate a causal role for striatal dopamine in gating, a key role for striatal dopamine in this output gating was suggested by ...*”.

20. It is difficult to gauge the effect sizes for the effects. Please add a discussion of this, perhaps in terms of the amount of variance explained.

The estimates for the effects that result from the Bayesian mixed-effects models of accuracy are on the logit scale. Exponentiating the logit scale estimate gives the odds ratio of an effect. For example, the main effect of methylphenidate on accuracy is 0.268 on the logit scale, which gives an odds ratio of 1.31. We have added this explanation in the methods section in line 1003: “*Note that the resulting effect estimates of response times are therefore on the log scale, and those for accuracy are on the logit scale. Exponentiating the logit scale estimate gives the odds ratio of an effect.*”

REVIEWERS' COMMENTS

Reviewer #1 (Remarks to the Author):

The authors have done a great job addressing my initial comments and suggestions. I have no further comments!

Reviewer #2 (Remarks to the Author):

The authors have done an admirable job addressing my critiques raised during the previous round. I have two remaining concerns that I would like to see addressed by the authors:

1. While it's great that the authors conducted additional analyses including age and session as covariates, it's not quite obvious to me how they did it. Reading between the lines I think the authors added age and session as additional main effects but not interaction terms. In either case, it would be helpful to include clearer description of what was done and tables containing coefficients pre- and post-controlling for age and session effects. All of this can be in the SI Materials but would be valuable for interested readers and future studies building on these findings.
2. I feel that Figure 7, which shows the schematic of key findings, could be improved in terms of both content and exposition. Content-wise, it would be helpful to explicitly indicate comparisons between drug and baseline, as well as making clearer which comparisons the authors are intending to highlight. In terms of exposition, the information to white space ratio is quite low for the figure, and may be easier to digest as a table in its current form.

Reviewer #3 (Remarks to the Author):

The revision has addressed most of the issues well and is clearer. Two small points remain-

Point 5- thanks for clarifying the randomisation and blinding process in the paper. please include in the manuscript.

Point 9: thanks for clarifying. the figure provides a reasonable justification for the cut-off. please add to the supplementary information so it is clear for readers

We thank the reviewers again for the time and effort they have invested in reviewing our work. We are happy to address their final comments on our revised manuscript below.

Our responses to the reviewers' points are in blue, with excerpts from the manuscript in italic. In the revised manuscript we have highlighted the changes in yellow. As instructed, the changes in the Supplementary Information are not highlighted.

We have changed all instances of "Supplemental Information" to "Supplementary Information" to adhere to the naming convention of Nature Communication (these changes are not highlighted). We have also decided to add a table of contents to the Supplementary Information document.

REVIEWERS' COMMENTS

Reviewer #1 (Remarks to the Author):

The authors have done a great job addressing my initial comments and suggestions. I have no further comments!

We are happy to hear that the modifications in response to the initial comments were satisfactory.

Reviewer #2 (Remarks to the Author):

The authors have done an admirable job addressing my critiques raised during the previous round. I have two remaining concerns that I would like to see addressed by the authors:

1. While it's great that the authors conducted additional analyses including age and session as covariates, it's not quite obvious to me how they did it. Reading between the lines I think the authors added age and session as additional main effects but not interaction terms. In either case, it would be helpful to include clearer description of what was done and tables containing coefficients pre- and post- controlling for age and session effects. All of this can be in the SI Materials but would be valuable for interested readers and future studies building on these findings.

The specifics of the additional models testing for the effects of age and session were indeed not clear enough. We included the factors age and session not only as main effects but also in interaction with the other factors. The factor session was included in both the fixed and random effects, while the between-participants variable age was only included in the fixed effects. We now included this information by changing lines 218-227 of the SI Materials to: *"To this end, we included age and session number as additional predictors in the Bayesian mixed-effects models, both as main effects as well as in interaction with the other factors. The factor session was included in both the fixed and random effects, while the between-participants variable age was only included in the fixed effects. These analyses revealed that age and session number did not affect the significant synthesis-dependent drug effects on reward versus punishment reversal accuracy (Table S4)."* We also changed the word "covariates" to "predictors" in line 262, since the word covariates might suggest that the factors were tested for main effects only.

We have included additional tables in the SI Results that show the model coefficients for the most important dopaminergic drug effects on accuracy and response times (Supplementary Table S4 and

Supplementary Table S5). These contain the model estimates from the original models, as well as the estimates from the models that included age and session number as confound variables.

2. I feel that Figure 7, which shows the schematic of key findings, could be improved in terms of both content and exposition. Content-wise, it would be helpful to explicitly indicate comparisons between drug and placebo, as well as making clearer which comparisons the authors are intending to highlight. In terms of exposition, the information to white space ratio is quite low for the figure, and may be easier to digest as a table in its current form.

Figure 7. Schematic of key findings. The dopaminergic drugs methylphenidate and sulpiride boosted outcome surprise signals in the caudate nucleus and stimulus-specific visual association cortex in 'low striatal dopamine' participants (top row panels). This was accompanied by the drugs boosting punishment compared with reward-based reversal learning in the 'low striatal dopamine' participants compared with the 'high striatal dopamine' participants (bottom left panel). By contrast, the drugs boosted relative reward-based reversal learning to a greater degree in 'high striatal dopamine' participants (bottom left panel), and methylphenidate also boosted the associated prefrontal BOLD signal to a greater degree in 'high striatal dopamine' participants (bottom right panel). MPH: methylphenidate; SUL: sulpiride.

Reviewer #3 (Remarks to the Author):

The revision has addressed most of the issues well and is clearer. Two small points remain-

Point 5- thanks for clarifying the randomisation and blinding process in the paper. please include in the manuscript.

We have adapted the sentence in line 576 to include this information: "The order was randomized by an independent researcher and the medication was prepared and coded by the pharmacy in accordance with the prespecified randomized order, ensuring that the experimenters as well as the participants were blind to the drug status on each session."

Point 9: thanks for clarifying. the figure provides a reasonable justification for the cut-off. please add to the supplementary information so it is clear for readers

In the relevant part of the methods section in the main text we now refer (line 655) to a newly added section in the Supplementary Information containing this information and the figure (lines 450-465; Supplementary Figure S15): "There was a minimal task performance criterion of having at least 20 reversals per session. Fewer than 20 reversal trials occurring in a session was considered a sign of task disengagement. To test this, we correlated the number of reversals in a session with the number of missed trials in that session, across all available sessions (i.e. without any exclusions). This revealed a strong negative correlation ($t_{(291)}=-17.49$, $p < 2.2e-16$), indicating that those sessions on which participants had few reversal trials also contained a large number of missed trials (rather than simply

incorrect trials), presumably reflecting task disengagement. Figure S15 illustrates this correlation and shows that the preset performance criterion of 20 reversals resulted in the exclusion of sessions with many missed trials.”